# South-American plate advance and forced Andean trench retreat as drivers for transient flat subduction episodes

Gerben Schepers[1], Douwe J.J. van Hinsbergen[1], Wim Spakman[1,2], Martha E. Kosters[1], Lydian M. Boschman[1] & Nadine McQuarrie[3]

At two trench segments below the Andes, the Nazca Plate is subducting sub-horizontally over ~200–300 km, thought to result from a combination of buoyant oceanic-plateau subduction and hydrodynamic mantle-wedge suction. Whether the actual conditions for both processes to work in concert existed is uncertain. Here we infer from a tectonic reconstruction of the Andes constructed in a mantle reference frame that the Nazca slab has retreated at ~2 cm per year since ~50 Ma. In the flat slab portions, no rollback has occurred since their formation at ~12 Ma, generating 'horse-shoe' slab geometries. We propose that, in concert with other drivers, an overpressured sub-slab mantle supporting the weight of the slab in an advancing upper plate-motion setting can locally impede rollback and maintain flat slabs until slab tearing releases the overpressure. Tear subduction re-establishes a continuous slab and allows the process to recur, providing a mechanism for the transient character of flat slabs.

[1] Department of Earth Sciences, Utrecht University, Heidelberglaan 2, 3584 CS Utrecht, The Netherlands. [2] Center for Earth Evolution and Dynamics (CEED), University of Oslo, Sem, Saelands vei 24, NO-0316 Oslo, Norway. [3] Geology and Planetary Science, University of Pittsburgh, 4107 O'Hara Street, Pittsburgh, Pennsylvania 15260, USA. Correspondence and requests for materials should be addressed to D.J.J.v.H. (email: D.J.J.vanHinsbergen@uu.nl).

With slab pull as major driver of subduction[1,2], the dynamic causes of the 200–300 km long 'Peruvian' and 'Pampean' flat slabs are intriguing[3–6] (Fig. 1). A key step towards assessing the possible causes of flat-slab generation is to reconstruct their kinematic evolutions based on geological expressions that date the onset of flat slab subduction. Such expressions include the migration of the volcanic front to the east away from the trench, a trench-normal expansion of arc magmatism, extinction of the magmatic arc, eastward migration of crustal deformation, and retro-foreland basin subsidence[7–9]. Apart from the Peruvian and the Pampean flat slabs that formed since ~11 and ~12 Ma, respectively[9,10], several older flat slab subduction episodes have been interpreted to have occurred in segments along the Andean subduction zone since ~42 Ma (ref. 9).

The Andean flat slabs were originally proposed to result from the overall positive buoyancy of the subducting Nazca plate induced by volcanism-related thickening along, for example, the Nazca and Juan Fernandez Ridges and a delayed basalt-to-eclogite transition in the subducted crust[4,11–16] (Fig. 1). For pre-Miocene flat slab episodes, however, the correlation between flat slab subduction and arrival of such positively buoyant bathymetric impactors at the trench are not supported by plate kinematic reconstructions[17], implying that other mechanisms, such as westward overriding plate motion relative to the mantle of the South American (SAM) plate, contribute to triggering flat slab subduction[15,16,18,19]. Recent numerical subduction modelling shows in addition that lateral changes in thermal thickness of the overriding plate[20] or a thick overriding lithosphere increase the propensity for creating upward hydrodynamic slab suction often expressed as an under-pressured mantle wedge[19,21,22], although this may not apply to the Pampean flat slab of Chile[22].

Whatever may be the cause, flat slab formation must comply with the relative motions between the subducting Nazca plate, the overriding SAM plate, and, importantly, the mantle. While the formation of the Andes has been viewed in context of absolute plate motion of the South American plate[23], that is, in a mantle frame of reference, flat-slab formation has not, even though it is essential for determining the role of, for example, slab rollback in the process. We will therefore first develop a kinematic reconstruction of shortening in the Andes. In combination with the absolute motion of South American trench estimated from mantle reference frames, this reconstruction will provide the absolute motion of the Andean trench. Next, with constraints on the age of flat slab formation (11–12 Ma)[9,10], the Nazca Plate subduction rate through time from the SAM-Antarctica(-Pacific)-Nazca plate circuit[24], the current dimensions of the flat slabs[5,6], and the absolute (African) plate motion evolution[25], we will quantify the relationship between the flat slab length, rollback of the slab bend, and the amount of subduction since the onset of flat slab formation. This will be used as basis to evaluate possible dynamics triggering and supporting flat slab subduction.

## Results

**Conceptual model.** We develop a mantle-fixed reconstruction since ~50 Ma, in which we restored the relative motions between the overriding SAM plate (O), the downgoing Nazca Plate (D), the intervening trench (T) and as a key new element, the slab bend (B) under the overriding plate that marks the end of the flat plate and slab where the downgoing slab dips into the upper mantle (M) (Fig. 2). In such reconstructions flat slabs appear when the westward retreat of the trench is faster than the westward rollback, which we define as the retreat of the slab bend relative to the mantle (Fig. 2b). The total amount of rollback

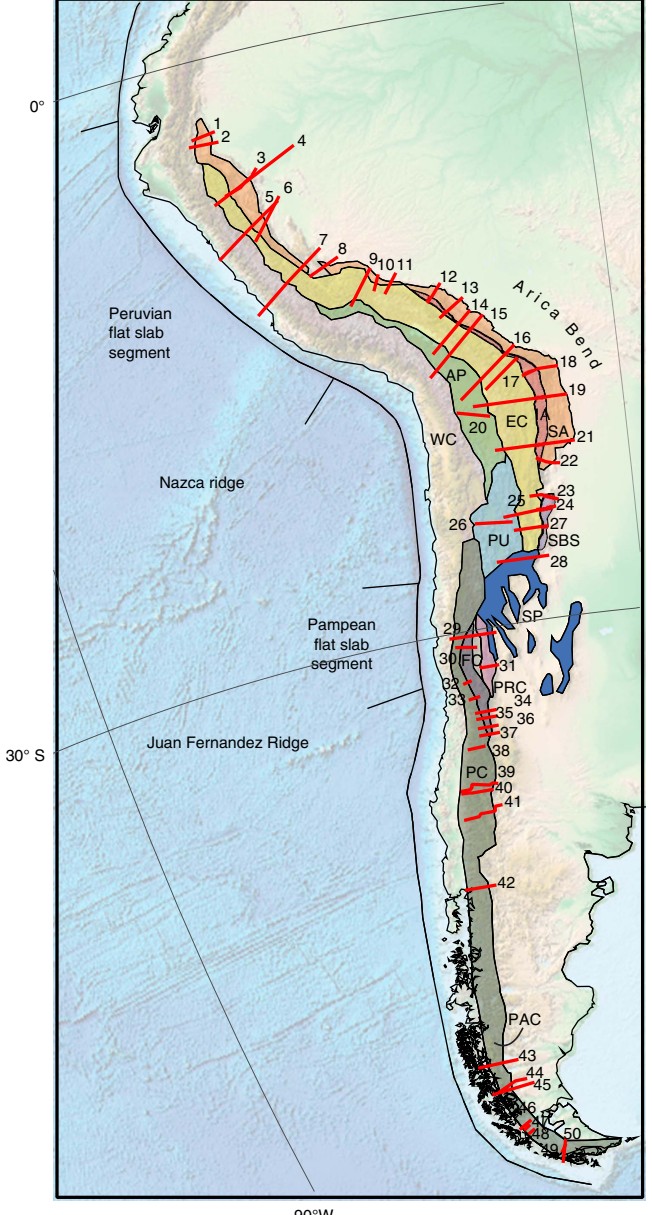

**Figure 1 | Tectonic map.** 3D Globe projection of the Andes[29], showing the boundaries of deforming geomorphologic regions, flat slab segments and ridges as discussed in this paper. AP, Altiplano Plateau; EC, Eastern Cordillera; FC, Frontal Cordillera; IA, Interandes; PAC, Patagonian Cordillera; PC, Principal Cordillera; PRC, Precordillera; PU, Puna Plateau; SA, Subandes; SBS, Santa Barbara System; SP, Sierras Pampeanas; WC, Western Cordillera. Numbered red lines indicate balanced cross sections; numbers refer to entries in Supplementary Table 1.

since the start of the reconstruction equals the total amount of SAM (O) motion relative to the mantle minus SAM shortening in the Andes, which we include in the trench motion (T; Fig. 2), and minus flat slab length (Mode 2; Fig. 2b). We assume that initially the slab bend coincided with the reconstructed position of the trench at 50 Ma. If flat slabs were already present at 50 Ma (ref. 26), the net amount of rollback would be even more than reconstructed here. In absence of rollback and overriding plate shortening (Fig. 2b.), at least 1,400 ± 180 km long flat slabs should have formed since ~50 Ma, corresponding to the total westward motion of SAM[25] (Fig. 2c). The modern

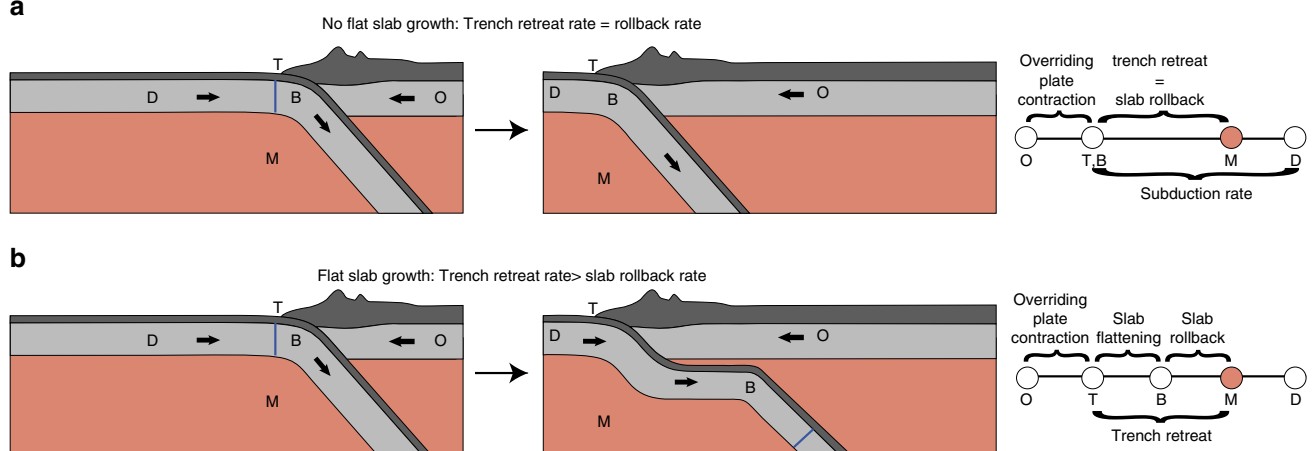

**Figure 2 | Conceptual model of flat slab flat slab subduction in a mantle reference frame.** We distinguish the kinematic vectors of the Overriding plate (O), the Downgoing plate (D), the Trench (T), the slab Bend (B) and the sub-lithospheric upper Mantle (M) in a 2-D trench-perpendicular cross section. Trench T is the intersection between the subduction thrust and the sea floor. Slab bend B is the hinge between the (sub-)horizontal and dipping slab segments. In cases of flat slab subduction B is separated from T by zone of low-angle subduction. Left, Middle panel: plates in a cross-sectional view at the onset and after a period of overriding plate advance, respectively. Right panel: plates and plate boundaries on a velocity line, where M is our reference, illustrating the following relationships: the difference between motion vectors of overriding and downgoing plate is the plate convergence rate (O minus D); of trench and downgoing plate is the subduction rate (T minus D); of trench and overriding plate is the overriding plate shortening rate (T minus O); of slab bend and trench is the flat slab formation rate (T minus B); of trench and mantle is the trench retreat rate (T minus M); and of slab bend and mantle is the slab rollback rate (B minus M). (**a**) subduction of D underneath O. T and B have the same relative velocity with respect to M, i.e., trench retreat equals slab rollback. The higher velocity of O relative to M is accommodated by overriding plate shortening. The blue line on the downgoing plate is a marker to indicate the lithosphere that is located at the trench at the onset of overriding plate advance. (**b**) represents subduction of a flat slab, whereby T has a higher relative velocity than B. In this example, both retreat relative to M. Trench retreat exceeds slab rollback, flat slab forms. The rate of subduction exceeds the rate of flat slab formation: the blue line passes both the position of the trench and the slab bend.

Andean flat slab segments are, however, only 200–300 km long. To assess the difference, we first need to quantify the overriding plate shortening history by kinematically restoring the Andean fold-thrust belt relative to SAM.

**Shortening reconstruction Andes.** Our shortening reconstruction of the Andean fold-thrust belt is based on 50 published balanced cross-sections (Fig. 3, Supplementary Table 1) distributed between 3°S and 56°S, combined with constraints on subduction erosion estimates, local strike-slip-fault displacement estimates and paleomagnetic data (see Methods section). Our restoration uses published maximum estimates of Andean shortening, which leads to a minimum estimate of trench retreat and hence reconstructs the minimum amount of roll-back required to avoid the formation of flat slabs. Shortening in the Andes is illustrated by subdividing the fold-thrust into 12 commonly recognized geomorphic regions (Figs 1 and 3), whereby we restored shortening identified in each zone. The Northern Andes, where another flat slab was recently reported[27], is not reconstructed in detail because of additional complexity due to interaction with a third (Caribbean) plate[28]. In the south, our reconstruction stops at the Scotia Sea.

Snapshots of the reconstruction of the Andes through time are shown in Fig. 3. The Bolivian Andes is the region with the largest total shortening, located in the centre of the Arica Bend. Our estimate is consistent with previous results[29] and estimates a shortening of ~420 km with rotations of 13° clockwise south of the bend and 13° counterclockwise north of the bend. To the south, shortening in the Argentinian Andes decreases to ~160 km. Approximately 150 km of shortening was determined for the northern part of our reconstructed region. Our reconstruction of Andean shortening, which benefits

from the largest number of balanced cross sections to date, coincides well with previous estimates of along-strike shortening variations in the Andes[30] (Fig. 4). Shortening models that do not take into account vertical axis rotations[23,31] yielded total maximum shortening estimates up to 100–200 km smaller than in our model, and this difference may be used as the first-order uncertainty on the reconstruction. Because the Andes form by shortening of the overriding plate above a subduction zone, we assume that the upper crustal shortening is representative for shortening of the entire lithosphere, in conjunction with earlier studies[32].

**Quantifying subduction and rollback.** Combining the shortening reconstruction of the Andes with the South America-Antarctica-(Pacific)-Nazca plate circuit[24], we arrive at the important result that ~1,000 km of Nazca plate has passed the trench and subducted since the 12 Ma onset of flat slab subduction (Fig. 5), whereas the modern flat slabs are only ~200–300 km long. By implication, the lithosphere that subducted at 12 Ma must therefore have moved through the slab bend and currently resides much deeper in the mantle while the observed flat slab only constitutes lithosphere that subducted since ~3 Ma.

This relationship between the length of subducted lithosphere since the onset of flat slab formation and the length of the flat slab suggests that the flat slabs may not be directly related to a resistance of the Nazca lithosphere against subduction. We therefore evaluate whether an alternative explanation may be found in the relationship between trench motion, slab bend motion, absolute SAM plate motion, and the observed flat slab by placing our reconstruction in a mantle reference frame. This will determine the amount of trench retreat relative to

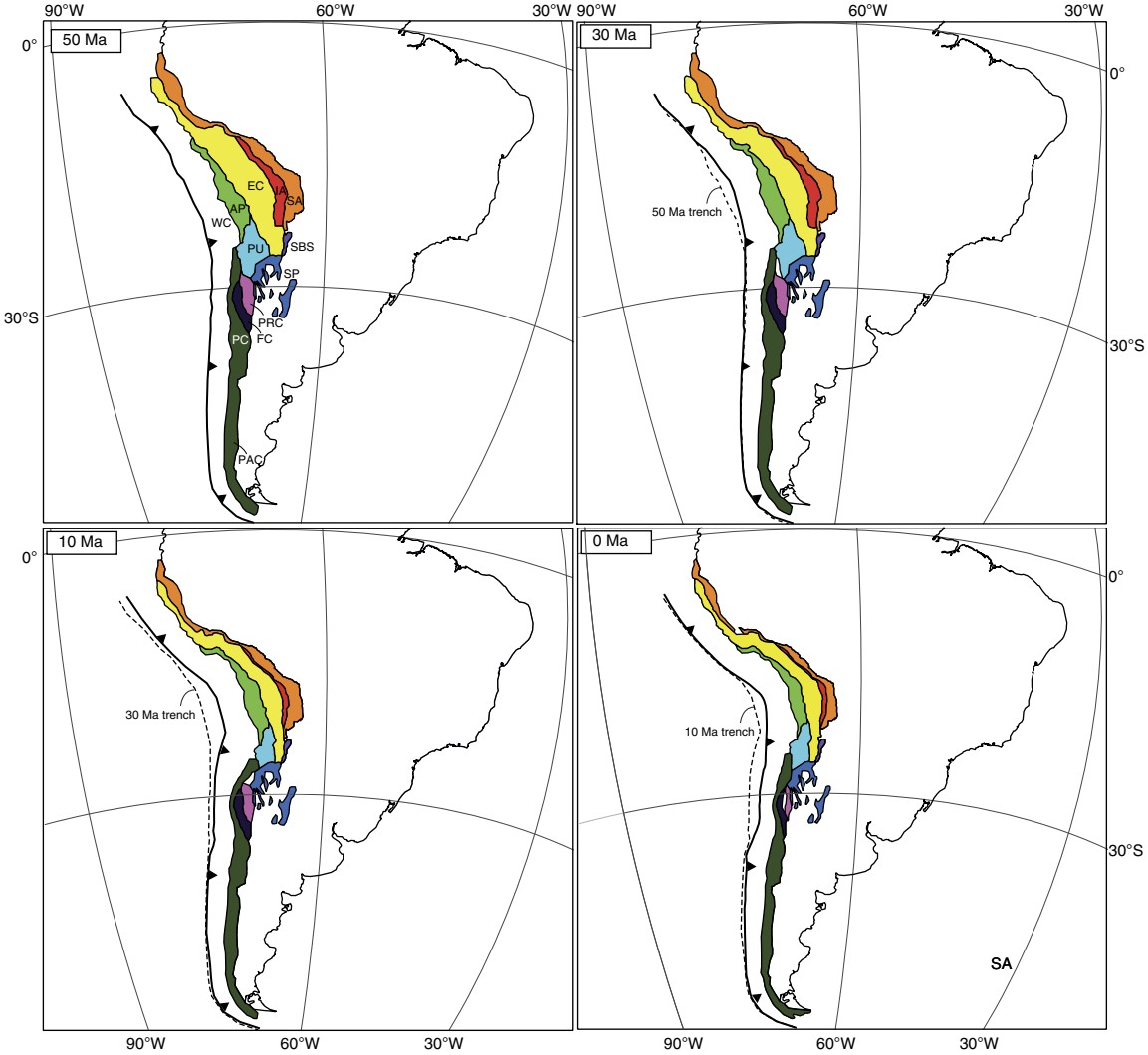

**Figure 3 | Reconstruction of the Andes and South American trench.** Reconstruction at at 50, 30, 10 and 0 Ma in a South America fixed frame. AP, Altiplano Plateau; EC, Eastern Cordillera; FC, Frontal Cordillera; IA, Interandes; PAC, Patagonian Cordillera.; PC, Principal Cordillera; PRC, Precordillera; PU, Puna Plateau; SA, Subandes; SBS, Santa Barbara System; SP, Sierras Pampeanas; WC, Western Cordillera.

the mantle since 50 Ma. A recent study[33] tested mantle reference frames against a set of independent geodynamic criteria of realistic subduction zone behaviour[34] and found that the global moving hotspot reference frame[25], two Indo-Atlantic moving hotspot reference frames[35,36], and a slab-fitted reference frame[37] perform best. Because the latter frame is subject to the largest, and poorest defined uncertainties and was designed particularly for Mesozoic absolute plate motions, we use the moving hotspot reference frames to place our reconstruction in absolute plate motion context. The global moving hotspot reference frame[25], shows ∼1,400 ± 180 km of westward absolute South American plate motion since 50 Ma. The Indo-Atlantic moving hotspot frame predict a motions that are ∼100 km more[35], or less[36] since this time (Fig. 6). Combined with our Andean reconstruction, we estimate a minimum of ∼1,000 ± 180 km of westward Andean trench retreat since 50 Ma, that is, with a long-term average trench retreat rate of ∼20 ± 4 mm per yr.

This long-term trench retreat rate implies that since ∼12 Ma, ∼240 ± 50 km of trench retreat occurred, which coincides with the width of the modern flat slabs. By implication, no significant rollback of the slab bend (that is, B and M coincide in the

velocity diagrams in Fig. 2b) has occurred in these segments since 12 Ma, while ∼1,000 km of the Nazca plate has continued to subduct. From a kinematic point of view, flat slab formation since ∼12 Ma resulted from SAM overriding the slab bend that remained essentially stationary in the mantle (Fig. 5).

## Discussion

We now explore a dynamic explanation for the stagnation of the slab bend relative to the mantle during flat slab formation. A thousand kilometres of trench retreat and without forming flat slabs of this length requires slab rollback. When viewed in a mantle reference frame, slab rollback requires that mantle material is removed from below the slab. This occurs through toroidal flow around slab edges, or through tears or holes in the slab, and poloidal flow around its base if possible[38–42]. The mantle flow speed and lateral length of the flow path in the sub-slab mantle limits the rollback speed and facilitates slab stagnation and potential flat slabs for laterally long subduction zones[43]. Slab stagnation may explain why most shortening, averaged over the last 50 Ma, occurred in the central segment of the Andes[29], a proposed potential rollback 'stagnation

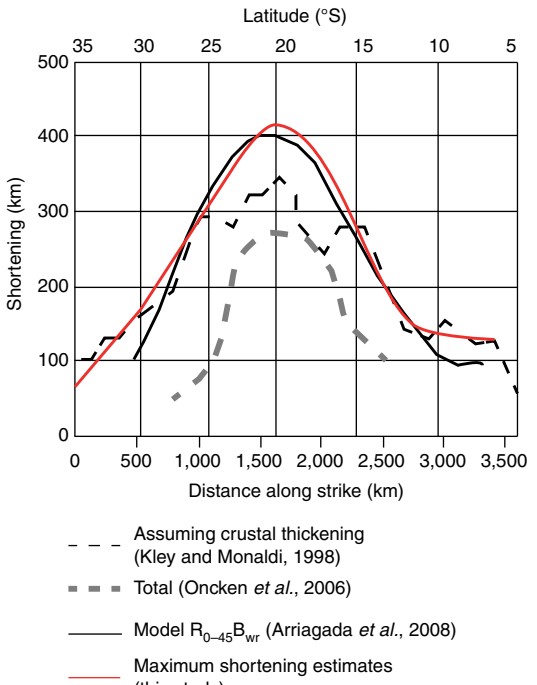

**Figure 4 | Shortening along the Central Andes.** Shortening estimated in various compilations (Modified from Arriagada et al.[30]).

point'[44] based on seismological inferences of trench-parallel mantle flow in the sub-slab mantle. Assuming a contiguous 7,000 km slab below the Andes, rollback would have been slowest in this central segment, and normal and shear stresses at the subduction interface between the advancing SAM and the downgoing Nazca plate would have been highest[43]. Thus flat slab formation is expected in the 50 Ma subduction history of the central Andes. However, the current slab geometry, with two flat slabs segments separated by a dipping slab that has been able to roll back rather than one flat segment in the central part of the Andean subduction zone suggests that the triggers and maintenance of flat slabs require additional mechanisms.

Our new kinematic restoration demonstrates that the explanation for the formation of the Andean flat slabs needs to take into account the overall setting of absolute plate motion-forced trench retreat and roll-back, and must allow for subducting lithosphere to move through the flat slab segment and slab bend. A proposed mechanism to explain the modern flat slabs includes subduction of a thick and buoyant oceanic crust (for example, an aseismic ridge), combined with a delayed basalt-to-eclogite transition for the negative buoyancy necessary for subduction at a steep angle[14,15,19] beyond the slab bend. A transition delay of ∼3 Myr would explain the ∼250 km long Pampean flat slab, provided that the subducted aseismic ridge was at least ∼1,000 km long. While subduction of aseismic ridges or otherwise buoyant oceanic crust may provide the initial trigger for a flat slab, other dynamic mechanisms may be called on to maintain the geometry and inhibit rollback. Recent numerical modelling[19,45] suggested that upward dynamic suction of the mantle wedge can only effectively lead to a local flat slab in case of the subduction of a buoyant oceanic plateau, although this effect depends on the adopted mantle rheology[18], and on the nature of the overriding plate, being more pronounced when the slab subducts under a thicker and stronger upper plate[46,47]. This dynamic suction as a mechanism of maintaining the Pampean flat slab is uncertain

due to the unknown proximity of the Amazonian craton[22]. Also it is debated if buoyant oceanic plateaus or ridges were available to initiate and maintain the Cenozoic flat slabs below the Andes[17].

Therefore we want to draw attention to the role of another potential dynamic contribution that may assist dynamic slab suction and does not require large amounts of subducted buoyant oceanic plateau to maintain flat slab subduction. This dynamic contribution is the creation of an overpressured sub-slab mantle as inferred in laboratory[48] and numerical experiments of subduction in an advancing upper plate setting[19]. An overpressured sub-slab mantle tends to lift the slab, may occur in concert with hydrodynamic slab suction above the slab[19,45,49] and is particularly reported for rollback of the Nazca slab[22,50] and of the Farallon slab being overridden in the Late Cretaceous and Paleogene by the North American plate[51]. The over-pressure of the sub-slab mantle drives trench-parallel mantle escape during rollback[43,50] and may even lead to slab stagnation for laterally wide slabs, where slowed removal of sub-slab mantle has become the rollback inhibiting process[43].

Our result that the slab bend of the flat slab did not appreciably roll back below the westward advancing SAM plate over the last 12 Myr confirms such slab stagnation for the flat slab regions of South America. It implies that mantle material below flattening segments of the slab does not escape and is in fact being trapped under the flattening slab by dipping slab segments to the North, East and South. This 'semi-dome', horse-shoe slab-geometry, open to the west (Fig. 7), may eventually develop toward a slab tunnel[52] that further supports flat subduction while allowing the eastward escape of material below the slab at the eastern tip of the semi-dome, or may collapse once the dynamic over-pressure of the sub-slab region is released by flow through slab tears[45,51].

We propound that in the case of a laterally long subduction zone with trench retreat forced by the advancing upper plate (as for the Nazca and Farallon subduction zones), an overpressured sub-slab mantle immediately responds to any local decrease in subduction angle, initiated by one or more of previously proposed flat slab triggers[4,11,15,17,19,21,53], to help create and maintain a flat slab that is instantaneously underlain by the overpressured sub-slab mantle. The trapped mantle underneath the flat slab would impede rollback of flat-slab segments, explaining our primary observation. Importantly, in the case of long subduction under a mantle-stationary overriding plate with large lithosphere thickness, 3-D numerical shows that local flat slab creation due to hydrodynamic slab suction alone is highly transient and short-lived (<5–6 Myr) because of sinking of slab to either side of the flat portion[47] or isolated by tearing directly around the flat slab[54]. We explain this as resulting from the absence of an overpressured sub-slab mantle, which we argue is naturally produced in a laterally long subduction zone undergoing trench retreat. The over-pressurized sub-slab mantle would contribute to longer flat slab support. Eventually, slab tearing at the edges of the flat slab, both observed[55,56] and modelled[14,54], or formation of slab tears/holes in the centre of the flat slab[22,45], will allow escape of the trapped mantle[56] releasing the sub-slab overpressure, and the flat slab segment (perhaps minus a part of its crust that underplates the upper plate[14]) steepens and becomes part of renewed slab rollback. Subsequent subducting of the slab tears can accommodate mantle throughput from the sub-slab mantle to the topside[51] until they disappear into the deep mantle. A possible example of a deep Nazca slab conduit, although of uncertain origin, facilitating such mantle flow can be found in recent tomography[57] of the Nazca slab under the Bolivian Orocline, Central Andes, where below depths of 600 km an eastward pointing cusp in slab geometry is identified that seems associated with a hole in the

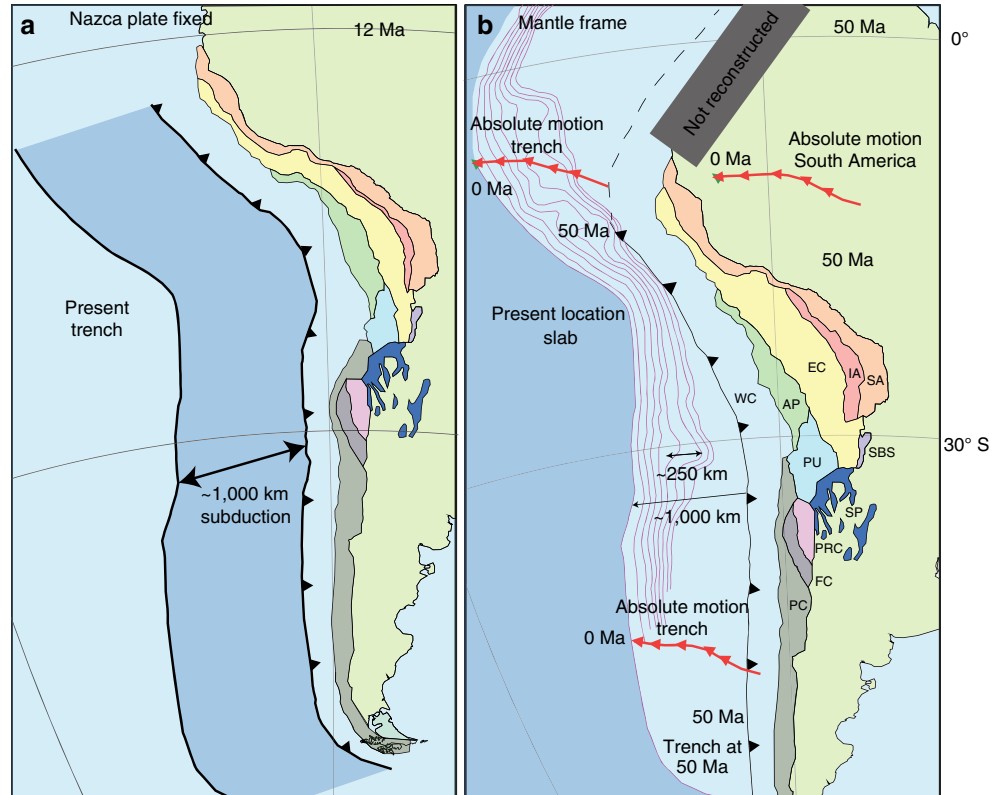

**Figure 5 | Reconstructions of the central and Southern Andean fold-and-thrust belt.** (**a**) Reconstruction of the Central and Southern Andean fold-and-thrust belt at 12 Ma with a fixed Nazca Plate colours of the deforming geomorphologic regions are the same as in Fig. 1. The distance between the present trench and the trench at 12 Ma equals the amount of subduction of the Nazca Plate underneath the South American Plate since 12 Ma, shown here in darker blue. (**b**) Reconstruction of the Central and Southern Andean fold-and-thrust belt at 50 Ma in a hotspot mantle reference frame[25], colours of the deforming geomorphologic regions are the same as in Fig. 1. The purple lines represent the present depth contour lines of the Nazca Plate from 0 to 160 km with a 20 km interval[6]. The amount of slab rollback for a dipping slab perpendicular to the present depth contour lines is ~1,000–1,100 km, for the flat slab segment this is ~250 km less. The arrow lines represent the motion paths[25] of one point on the trench and the South American Plate since 50 Ma to its present position in steps of 5 Myr. WC, Western Cordillera; AP, Altiplano Plateau; EC, Eastern Cordillera; IA, Interandes; SA, Subandes; PU, Puna Plateau; SBS, Santa Barbara System; PC, Principal Cordillera; FC, Frontal Cordillera; PRC, Precordillera; SP, Sierras Pampeanas; PAC, Patagonian Cordillera.

slab. This deep tear in the slab may perhaps have a geometrical (lateral slab bending) origin as it also occurs as such in recent 3-D modelling of the Nazca subduction[45]. An overview of SKS splitting results[58] shows anomalous and puzzling E-W fast splitting patterns in the subslab mantle[59] in a narrow zone (~150 km in N-S extent) that geographically aligns in E-W direction with the deep slab hole. Shear deformation associated with upper mantle flow toward the hole and escaping eastward trough the hole could provide a plausible explanation for the observed splitting observations while the flow itself would release subslab pressure and contribute to local rollback of the Central Andes slab.

With reconstructed Cenozoic subduction rates of the Nazca Plate (80–90 km Myr$^{-1}$) the slab tears that ended previous periods of flat subduction have rapidly disappeared in the deeper upper mantle followed by subduction of a wide sheet of continuous slab, which sets the stage for a next phase of flat subduction (partly) maintained by the overpressured sub-slab mantle (Fig. 7). This mechanism facilitates flat slabs that are both transient and regional features as suggested from the geological record[9]. Our kinematic analysis provides first-order constraints on the long-debated formation of the Andean subduction zone flat slab segments: we show for the example of the present-day flat slabs that their length coincides with the amount of westward absolute Andean trench motion since

the 12 Ma onset of flat slab formation. From this, we draw the conclusion that the slab bend has been essentially stationary in the mantle, and rollback was apparently impeded. We suggest that previous stages of flat slab subduction below the Andes, inferred from magmatic and structural records[9], can be explained in a similar way. We foresee that similar first-order constraints from tectonic reconstruction in a mantle frame may be obtained on the evolution of the long debated Late Cretaceous to Eocene Laramide flat slab in western North America[60] which is similarly characterized by subduction rates of 80–90 km Myr$^{-1}$ (ref. 24), and absolute westward motion of the North American plate[25,37] allowing for rapid subduction of tears and subsequent build-up of sub-slab pressure. Our research highlights that whereas the kinematic evolution of mountain belts is straightforwardly analysed using a relative plate circuit, it is of key importance to study the geodynamics of subduction zones in a mantle reference frame such that the coupling between plates, subducting slabs and ambient mantle is properly accounted for.

## Methods

**Reconstruction approach.** Our reconstruction is made using the freely available software Gplates (http://www.gplates.org)[61]. We provide the shape and rotation file of the reconstruction as Supplementary Data 1 and 2, respectively. We illustrated the shortening partitioning over the Andes using 12 commonly defined

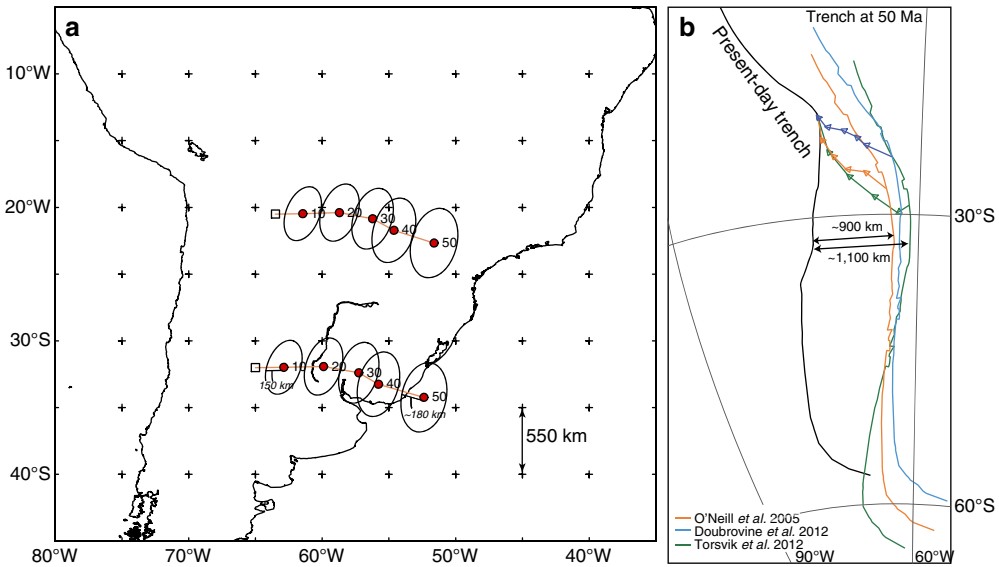

**Figure 6 | Absolute plate motions and trench retreat estimates.** (**a**) Absolute plate motion track of South America estimated from the global moving hotspot reference frame of Doubrovine et al.[25], including error bars. These estimates lie at the basis of our analysis of Andean trench retreat since 50 Ma. (**b**) Absolute trench retreat estimates, based on combining our estimate of Andean shortening with absolute plate motion models for South America, based on the moving hotspot reference frames of Doubrovine et al.[25] (preferred), Torsvik et al.[35] and O'Neill et al.[36]. The latter two frames would predict 100 km higher or lower trench retreat, within error of the Doubrovine et al.[25] frame.

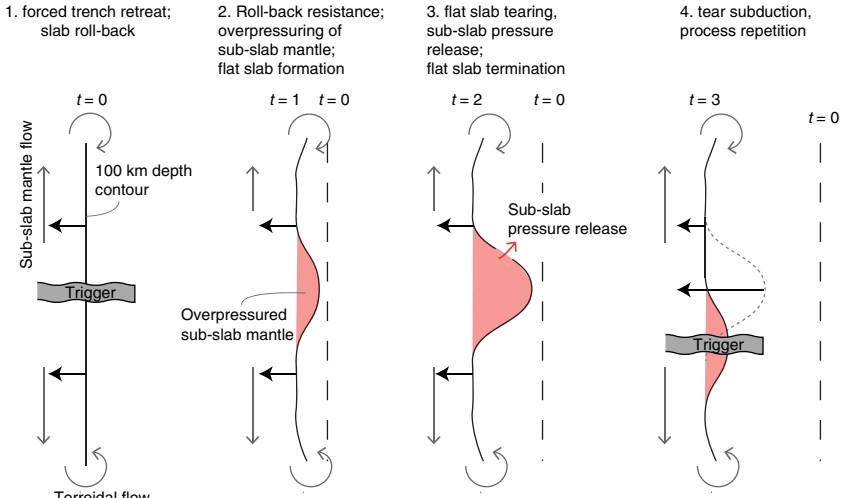

**Figure 7 | Conceptual model of the formation of overpressured sub-slab mantle dynamically supporting flat slabs in a top view of the 100 km depth contour line.** Stage 1 represents the rollback of a slab with trench-parallel mantle flow, toroidal flow and the arrival of a triggering mechanism (for example, arrival of more buoyanty material at the trench) for a decreased subduction angle, for example, a more buoyant body of lithosphere. Stage 2: mantle trapped below the flatter subduction segment in the overall context of slab rollback is trapped and becomes overpressured dynamically supporting a flat slab segment. Stage 3: the stresses on the flat slab segment induce tearing and sub-slab overpressure is released; flat slab roll-back occurs. Stage 4: slab segment with the slab tear progressively subducts to deeper levels and is replaced at shallower levels by slab material that is continuous.

geomorphologic regions, which are represented by the areas between different polylines in the reconstruction. Each polyline is subdivided in smaller portions based on the locations of cross-sections. These lines move with respect to each other during deformation and change in the area in between them reflects documented shortening. We reconstruct Andean deformation with respect to a fixed non-deforming South American continent in the east.

**Constraints used for reconstruction.** The kinematic reconstruction of the Andes is primarily based on shortening estimates from a total of 50 published balanced cross sections. Shortening directions have been reconstructed parallel to balanced sections where no vertical axis rotations were documented. For the timing of deformation, we followed the interpretations of the authors of the balanced cross sections, who generally constrained timing from stratigraphic ages of basin deposits, in combination with thermochronology from $^{40}$Ar/$^{39}$Ar of micas and

feldspars, zircon fission track, zircon (U-Th)/He, apatite fission track and apatite (U-Th)/He analyses. These indicate that Andean deformation occurred mainly in the last 50 Ma (ref. 29). Paleomagnetically documented rotation patterns in the Central Andes reveal counterclockwise rotation to the North, and clockwise rotations to the South of the Arica bend. Andean paleomagnetic data were primarily derived from volcanic forearc rocks and not from the retroarc fold-thrust belt where most deformation occurred since[30] and thus reflect the total rotation differences accommodated through regional shortening variations in the Andean fold-thrust belt. Displacements along strike-slip faults, finally, are used where well constrained. For instance, geological and seismologic data indicate that the Cochabamba and the Rio Novillero Faults are left-lateral and right-lateral respectively, although the amount of displacement is poorly constrained[29]. The displacement and timing along the Magallanes–Fagnano fault system are also still under debate. The proposed displacements are in the same order of magnitude (20–55 km) but the age of formation ranges from 100 to 7 Ma (ref. 62).

**Data availability.** The Gplates rotation and shape files, as well as a table of compiled shortening estimates and their literature sources that come with the Andes reconstruction are provided as Supplementary Files to this paper.

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

## Acknowledgements

D.J.J.v.H. acknowledges ERC Starting Grant 306810 (SINK) and NWO Vidi grant 864.11.004. L.M.B. acknowledges NWO grant 824.01.004. W.S. acknowledges support from the Research Council of Norway through its Centres of Excellence funding scheme, project number 223272. We appreciated constructive comments of Wouter Schellart and two anonymous reviewers.

## Author contributions

G.S., D.J.J.v.H., M.E.K., L.M.B., N.M. developed reconstruction. D.J.J.v.H. and W.S. placed reconstruction in geodynamic context. G.S., D.J.J.v.H., W.S. wrote the manuscript.

## Additional information

**Competing interests:** The authors declare no competing financial interests.

