## [Peer Review File · Nature Communications]

Reviewers' Comments:

Reviewer #1 (Remarks to the Author)

This manuscript proposes a conceptual model for the development of a flat (subhorizontal) subducting plate: in a subduction zone where upper plate advance is faster than slab rollback, a flat slab may develop. The manuscript emphasizes the role of high pressures below the slab in the flattening process. Flat slab removal is proposed to be triggered by a slab tear or breakage, which reduces the subslab pressure, allowing the slab to rollback to a steep angle. The history of the South America subduction zone is used to support this model, and the manuscript provides a new assessment of shortening of the western side of South America and motions of the oceanic and continental plates and trench relative to the deep mantle. I can not comment in detail about the reconstructions of South America plate motions and deformation, so my review will focus on the conceptual model.

Overall, this is an interesting paper that would be of broad interest to those studying subduction zones and mantle dynamics. It highlights the importance of the plate motions (with respect to the deep mantle) in controlling slab dynamics. Such plate motions are included in geodynamic models of flat subduction, especially recent models of South America (e.g, Manea et al., 2012; Hu et al, 2016; Hu and Liu, 2016). I feel that these papers do not emphasize this effect enough; this point is buried within the discussions of complex rheologies and features in the upper and lower subducting plate. It is refreshing to read a manuscript that strips things back to first-order considerations. The hypothesis is nicely presented and provides a testable idea that could be examined in other subduction zones with flat slabs. I like the simplicity of the proposed model. My main concern is about the novelty of this work. The proposed model may not be a significant step forward from existing work, although it is presented in a clearer and more cohesive fashion.

As mentioned, the effects of plate motions have already been demonstrated in previous geodynamic studies. In particular, van Hunen et al. (GRL, 2002; doi 10.1029/2001GL014004) specifically investigate the role of continental motion and oceanic plateau subduction for South America. They conclude that plate motion (in an absolute reference frame) is most important for generating a flat slab. It is surprising that this paper is not referenced in the current manuscript. The high sub-slab pressure is a direct consequence of the plate motions and thus, it is also included in the the previous geodynamic models, although it is not typically discussed explicitly. I think that the originality of this point is a little over-stated in the current manuscript. Overall, the presentation of the proposed model would be strengthened by drawing on these earlier numerical studies (van Hunen, Hu, etc.) to provide quantitative information about the magnitude of subslab pressures, etc.

The discussion of flat slab removal says that tears that ended earlier flat slabs have now disappeared into the deeper upper mantle (Lines 128-130). It would be helpful to have a more detailed discussion about observations that support the proposed model, such as tomographic images, geological features suggesting slab tears (e.g., adakitic magmas; Hu and Liu, 2016). Right now, this aspect of the model comes across as fairly ambiguous and unsupported.

Other minor comments:

- in the abstract, the meaning of "horse-shoe slab geometry" (Line 10) is not clear. I did not understand this until after reading the manuscript.

- Lines 76-80 – there should be a reference to the initial Russo and Silver studies that discuss slab movement and mantle flow (Science 1994; Geology 1996)

- Figure 2 – what are Stage 1 and Stage 2? Seems like these don't correspond with the Stages in

the Figure 3 caption.

- also on Figure 2 - I don't understand the vertical blue line. The caption says that it is a locally thickened part of the plate. The figure suggests this is a very thin region, but oceanic plateaus are typically hundreds of km wide (Line 99 suggests a width of 1000 km).

Reviewer #2 (Remarks to the Author)

Review of "South-America plate advance and forced Andean trench retreat as drivers for transient flat subduction episodes"

This manuscript presents a reconstruction of the South American subduction zone for the last ~50 Myr and discusses a conceptual model to provide new insight into the process of flat slab subduction, a process in which a significant portion (several hundred km) of the subducted slab lies (sub-)horizontally below the leading edge of the overriding plate. The reconstruction presented in Fig. 1 and the cartoons presented in Fig. 2 very nicely illustrate and explain (both for South America and in a generic sense) how flat slabs can grow in a kinematic framework, where oceanward retreat of the trench is faster than oceanward migration of the slab hinge at depth (i.e. the bending zone that connects the flat slab segment and the deeper (and steeper) portion of the slab).

The paper is very topical and brings important new insight and kinematic constraints to the debate on the origin of flat slab subduction, which is currently a hot topic in the literature. There are a number of (relatively minor) issues and comments that need to be addressed, most importantly related to the significance of sub-slab overpressure and the choice of "absolute" reference frame (see below). Once these are addressed, I would recommend that the paper is accepted for publication.

Best wishes,

Wouter Schellart

Main comments:

On L115-118 and L122-127 you argue that a buoyancy-driven change (reduction) in subduction dip angle creates an overpressure in the sub-slab mantle region, which then creates and sustains the flat slab segment. However, one would think that a buoyancy-driven reduction in slab dip angle would cause an underpressure in the sub-slab domain (as the slab is lifted upward, thereby reducing the pressure in the mantle below the slab). As such, the origin of sub-slab overpressure is not entirely clear and requires some more discussion and explanation. (note also that you write both "overpressure" and "over-pressure").

Reconstruction in Fig. 2c: This reconstruction uses the hotspot reference frame of Doubrovine et al. [2012]. Why do you choose this reference frame? You have to provide some justification in the main text (or refer to a justification that you provide in the SI). At present, you provide no justification for this choice, while it is generally known that different choices of reference frame (e.g. Pacific hotspot of Gripp and Gordon [2002], Indo-Atlantic hotspot of Muller et al. [1992] or O'Neill et al. [2005], global hotspot [e.g. Gordon and Jurdy, 1986; Doubrovine et al., 2012], no-net-rotation reference frames, etc.) can provide substantially different plate velocities and trench velocities [e.g. Funiciello et al., 2008; Schellart et al., 2008; Shephard et al., 2012; Williams et al., 2015]. As such, the choice of your reference frame affects the numerical outcomes of your kinematic reconstruction (e.g. trench migration distance, trench velocity, rollback distance, etc.). You illustrate this in Fig. S5, but this is for three rather similar reference frames. For example, what would be the trench migration and trench velocity in the (very different) Pacific hotspot

reference frame of Gripp and Gordon [2002]?

I personally favour Indo-Atlantic hotspot reference frames as, for example, there are indications that the Indo-Atlantic hotspot reference frames (e.g. from O'Neill et al. [2005]) work very well as a subduction zone/deep mantle reference frame, because they are most in agreement in scenarios when surface observations and plate tectonic reconstructions are tied with observed deep mantle slab anomalies [e.g. Schellart and Spakman, 2015]. Additionally, they are most in agreement when combining reference-frame dependent subduction kinematics, dynamic subduction modelling and observed slab geometries [e.g. Schellart, 2011], and agree with V-shaped structures of fracture zone segments linking across spreading ridges in the Atlantic Ocean [Muller and Roest, 1992].

In any case, some discussion and justification is required.

Other comments:

L20 and L22: "towards the overriding plate" This is a bit ambiguous, as the volcanic front and deformation are already located in the overriding plate. Maybe you could say something like "away from the trench"?

L74-75: "the observed flat slab only constitutes the last ~3 Myr of subduction." I would replace this with something like "the observed flat slab segment only constitutes the last ~3 Myr of subducted oceanic lithosphere." to be more precise.

L82: "... friction would have been highest." This has not much to do with friction coefficients. It would be more accurate to say "... normal and shear stresses at the subduction zone interface would have been highest".

L122-127 and elsewhere. One of the general conclusions, that flat slab subduction is caused by both forced trenchward overriding plate motion, which forces trench retreat, and subduction of buoyant features, is essentially the same as that of van Hunen et al. [2002], who base their conclusion on numerical subduction models of progressive subduction in 2D space. I think this work should be cited in the text.

L164: "upper mantle" Be more precise: this is the sub-lithospheric upper mantle.

L165: "cross section trench-perpendicular" Replace with "trench-perpendicular cross section"

L166-167: "the slab bend as the hinge between the (sub-)horizontal and dipping portions of the subducting plate"

This definition would actually indicate the bend at the trench, so you need to be more precise, e.g. "the hinge between the (sub-)horizontal slab segment underlying the overriding plate and the dipping slab segment further down the subduction zone"

L173 and L183: Replace "contraction" with "shortening"

L199: "Slab healing" is not the right choice of words here, as it would imply that at the place of the slab tear, this tear "heals" and so disappears, making the slab continuous again. This is not what you mean here. You mean that the slab segment with the slab tear progressively subducts to deeper levels and is replaced at shallower levels by slab material that is continuous. Anyway, I suggest to rephrase to be more precise and avoid possible confusion.

In Fig. 3, for stages $t=1$, $t=2$ and $t=3$, the curvature of the slab edges (convex towards the mantle wedge side) is the wrong way. Toroidal mantle return flow at the lateral edges of a wide slab (and narrower slabs for that matter) forces these lateral slab edges to attain a curvature that is concave

towards the mantle wedge side. See for example Fig. 4 in Schellart et al. [2007] and Fig. 4 in Morra et al. [2006].

Reviewer #3

General Review Questions:

1. Who will be interested in reading the paper, and why?

This is an exciting paper that will be of great interest to the broader geoscience community, including the subdisciplines of geology, structural geology, tectonics, sedimentology and stratigraphy, geophysics (paleomagnetism and seismology), and geodynamics. The paper would reach a broad audience because of the use of a range of data types and the application of these data to addressing the general and not fully understood tectonic process of flat slab subduction.

2. What are the main claims of the paper and how significant are they?

The paper claims that it can connect the width of the portion of the subducting plate that is undergoing flat slab subduction to the relative motion between the trench and the rollback of the distal bend in the flat slab. The paper suggests the mechanism to maintain the flat slab is the overpressured sub-slab mantle, that has been trapped due to a particular 3D slab configuration that prevents the subslab mantle from escaping. Eventually, the slab tears and the overpressured mantle can escape, reducing subslab support on the slab and allowing the slab to steepen. Thus, this is a time-dependent phenomenon that can explain the changes in flat slab geometry through time.

In addition, the paper presents a synthesis of geologic data used to construct estimates of tectonic shortening along the South American subduction margin. The shortening estimates are compared to that by previous studies and show an along strike variation in shortening (see supplementary figure). The shortening data are used in the tectonic reconstruction of the upper plate. In addition, the data may be used to constrain where the distal bend of the slab was located in the subsurface, which is in turn used to constrain the width of the flat slab segment through time. However, this is not clear. Uplift constraints and volcanism are also used to constrain the paleo-geometry and timing of changes in slab morphology.

The geodynamic implications are very significant, and put forward a newly formed perspective on flat slab subduction processes very closely tied to observational data. The data compilation presented in the supplemental information is quite extensive. It by itself will likely be used in future studies.

3. Is the paper likely to be one of the five most significant papers published in the discipline this year?

It is not clear if it will be in the top 5, but it certainly will raise quite an interest and probably direct multiple new lines of inquiry in many subdisciplines in geoscience. In other words, it has the potential to be a very significant paper and have a large impact on the field.

4. How does the paper stand out from others in its field?

The paper stands out in terms of the synthesis of data and the integration of geologic observational data into a tectonic reconstruction as well as to a geodynamic framework that is closely tied to recent results from studies in computational geodynamics.

5. Are the claims novel? If not, which published papers compromise novelty?

The geodynamic mechanism has been recently suggested in several papers cited in the paper. However, the other papers do not make the connections to the geologic data and the tectonic reconstructions in the way and the extent that is presented in this paper.

6. Are the claims convincing? If not, what further evidence is needed?

There is likely sufficient evidence to support the claims. However, in places the paper could benefit from clarification of what the specific evidence are and how they are used in the paper.

For example, are the thin-skinned deformation constraints being applied to constrain whole-scale lithospheric shortening and previous plate length? If so, the claims could be strengthened by clarification of how thin-skinned crustal deformation is used to constrain deformation of the entire plate (mantle lithosphere as well as crust).

In addition, what exactly is used to identify where the slab was flat and to identify the distance from the trench at which the bend in the flat slab occurred in the past? For example, recent geodynamic models show the distal bend in a flat slab often correlates with a basin in the upper plate, e.g., Jadamec et al. (EPSL 2013) and Eakin et al. (EPSL 2014).

The relative motions and importance of reference frames are emphasized, but they could be better logically connected within the text. Figures 1 and 2 are the opportunity to do this, but could be better presented. Clarify why the different reference frames are used for Figure 1b,c. Figure 1 in general is hard to follow. In the reconstructions, is the trench geometry fixed? Does it curve through time as in Schellart et al. (Nature, 2007), and what are the implications of a time evolving plate boundary shape, e.g. Moresi et al. (Nature, 2015), on the reconstruction length estimates?

Clarify the velocity diagrams in Figure 2, as well as the subduction modes and their labeling.

What is Figure 3 based on? Is this from a geodynamic model result from the authors or elsewhere in the literature? Is it from the flat slab width determined from geologic estimates through time? Figure 3 is a good attempt to use simplicity to convey the concepts, but the parts are not clearly labeled. For example, what is the gray swiggly band in the left diagram? Is that the more buoyant body?

Also, according to Gutscher et al. (Tectonics, 2000), for example, the length of the modern flat slab segments beneath South America can be quite variable in some regions are greater than 400 km. This longer length is not really addressed. Can the authors comment?

7. Are there other experiments or work that would strengthen the paper further?

There is a large data compilation. How is it used could be better clarified. Including a geodynamic model result in the supplemental figure would be useful to demonstrate the slab pressure, and the time evolution of the slab pressure changes and slab dip changes, if the authors have a figure from previous work.

8. How much would further work improve it, and how difficult would this be? Would it take a long time?

I think it would be in the scope of a minor to major revision. Not every reader may agree with the conclusions, but if the assumptions are more clearly laid out, then the reader could better assess the results.

9. Are the claims appropriately discussed in the context of previous literature?

There is actually a very good overview of the relevant literature. The authors have done a nice job on this, particularly so, with the breadth of the paper.

In addition to the papers above, it would be useful to include (a) Schellart (JGR, 2004) in the discussion of toroidal and poloidal flow (lines 77-78) and (b) Jadamec and Billen (Nature, 2010) and Stadler et al. (Science, 2010) in the discussion in lines 78-79, as the upper mantle viscosity plays a major role in determining the mantle flow speed. In addition, in linking the mantle flow to upper plate shortening and deformation, these two papers are relevant (Capitanio et al. (Nature, 2011) and Schellart and Moresi (JGR 2013)). This paragraph in general, lines 76-84, is unclear.

Mason et al. (Tectonophysics, 2010) is relevant for lines 119-120. For line 119, is the tearing of the edge of the slab a lateral edge as in Jadamec and Billen (Nature, 2010) or at the base of the slab (Arrial and Billen)?

In lines 33-36, it would be useful to include other models that show the effect of the upper plate on flat slab subduction, that suggest additional mechanisms, e.g., Sharples et al. (JGR, 2014) and Rodríguez-González et al. (EPSL, 2014).

10. If the manuscript is unacceptable, is the study sufficiently promising to encourage the authors to resubmit?

This is a very interesting paper, and I recommend major revision.

11. If the manuscript is unacceptable but promising, what specific work is needed to make it acceptable?

Additional Questions to Consider:

1. Is the manuscript clearly written? If not, how could it be made more clear or accessible to nonspecialists?

The paper is clearly written. There could be some tightening in terms of the logic in linking how the observations, geodynamic modeling results, and choice of reference frames all lead to the main result.

2. Would readers outside the discipline benefit from a schematic of the main result to accompany publication?

Yes, but I think this is what Figure 3 is aiming to do. Revision of Figure 3 could achieve this.

3. Could the manuscript be shortened? (Because of pressure on space in our printed pages we aim to publish manuscripts as short as is consistent with a persuasive message.)

Yes, the paper could be tightened a little in terms of streamlining the logic. This would likely not save more than 1 paragraph though.

4. Should the authors be asked to provide supplementary methods or data to accompany the paper online? (Such data might include source code for modelling studies, detailed experimental protocols or mathematical derivations.)

More information on the tectonic reconstructions in terms of the upper plate length and subducting plate paleo-geometry would strengthen the paper.

5. Have the authors done themselves justice without overselling their claims?

I think so. Addressing the comments elsewhere in the review and including the broader scope of geodynamic modeling studies would do the paper more justice, but expanding the perspective.

6. Have they been fair in their treatment of previous literature?

Yes, the authors have done a good job. Including several additional papers would give a more balanced discussion.

7. Have they provided sufficient methodological detail that the experiments could be reproduced?

Yes, but again, more detail on the reconstructions and what exactly was used, and what points were measured to obtain slab length, would be strengthen the paper.

8. Is the statistical analysis of the data sound, and does it conform to the journal's guidelines?

There are no error bars on the individual shortening estimates, but this may be hard to constrain. In comparing the different data sets, it is not clear how to systematically weigh the studies if there were to be significant error ranges between studies. It would be useful to show a plot of the slab length through time and what data were used to constrain that length, and for what parts of the subduction zone. Figure 1 may be trying to do this, but what geologic data from the surface determines the location of the bend in the slab at depth?

9. Are the reagents generally available?

I don't the reagents question applies to this paper. Chemical experiments were not being conducted.

10. Are there any special ethical concerns arising from the use of human or other animal subjects?

No, there are no special ethical concerns regarding human or other animal subjects.

Reviewers' comments:

Reviewer #1 (Remarks to the Author):

This manuscript proposes a conceptual model for the development of a flat (subhorizontal) subducting plate: in a subduction zone where upper plate advance is faster than slab rollback, a flat slab may develop. The manuscript emphasizes the role of high pressures below the slab in the flattening process. Flat slab removal is proposed to be triggered by a slab tear or breakage, which reduces the subslab pressure, allowing the slab to rollback to a steep angle. The history of the South America subduction zone is used to support this model, and the manuscript provides a new assessment of shortening of the western side of South America and motions of the oceanic and continental plates and trench relative to the deep mantle. I can not comment in detail about the reconstructions of South America plate motions and deformation, so my review will focus on the conceptual model.

Overall, this is an interesting paper that would be of broad interest to those studying subduction zones and mantle dynamics. It highlights the importance of the plate motions (with respect to the deep mantle) in controlling slab dynamics. Such plate motions are included in geodynamic models of flat subduction, especially recent

models of South America (e.g, Manea et al., 2012; Hu et al, 2016; Hu and Liu, 2016). I feel that these papers do not emphasize this effect enough; this point is buried within the discussions of complex rheologies and features in the upper and lower subducting plate. It is refreshing to read a manuscript that strips things back to first-order considerations. The hypothesis is nicely presented and provides a testable idea that could be examined in other subduction zones with flat slabs. I like the simplicity of the proposed model. My main concern is about the novelty of this work. The proposed model may not be a significant step forward from existing work, although it is presented in a clearer and more cohesive fashion.

The novelty of this work is two-fold: First, we draw specific attention to the still under appraised possibility that an overpressured sub-slab mantle may help sustain flat subduction independent of the subduction of buoyant material, particularly in the case of laterally wide subduction zones as the Andes. Combined with observed slab tearing (Antonijevic et al. 2015 & 2016) leads us to propose a new view on the transient nature of flat slab episodes. We arrive at this proposition because of our important finding that no rollback occurred at the location of the flat slabs. The latter conclusion resulted from our second innovation which is providing the first Andes-wide plate reconstruction of Andean deformation that allows us to quantify the role of absolute plate motion on trench migration.

As mentioned, the effects of plate motions have already been demonstrated in previous geodynamic studies. In particular, van Hunen et al. (GRL, 2002; doi 10.1029/2001GL014004) specifically investigate the role of continental motion and oceanic plateau subduction for South America. They conclude that plate motion (in an absolute reference frame) is most important for generating a flat slab. It is surprising that this paper is not referenced in the current manuscript.

Indeed, this reference is an important omission which we actually cannot explain since we know his work well. We have also incorporated this reference and one other: van Hunen et al. 2004. In addition we also included a reference to the most recent study on causes of flat slab subduction which also considers plate motion (Huangfu et a. 2016)

The high sub-slab pressure is a direct consequence of the plate motions and thus, it is also included in the the previous geodynamic models, although it is not typically discussed explicitly. I think that the originality of this point is a little over-stated in the current manuscript. Overall, the presentation of the proposed model would be strengthened by drawing on these earlier numerical studies (van Hunen, Hu, etc.) to provide quantitative information about the magnitude of subslab pressures, etc.

We added more text on sub-slab pressure based on the few modeling papers that explicitly mention this (although not recognize overpressure as an important driver for sustaining flat subduction; which we do here). These papers are not explicit on the magnitude of modeled pressures and we do not dare to comment on what would be needed. This needs to be further investigated in numerical modeling dedicated to the special setting of the Nazca subduction.

We changed/added (new references in full): *“This dynamic contribution concerns the creation of an over-pressured sub-slab mantle as is inferred in laboratory⁴⁹ and numerical experiments of subduction that assumed a mantle-stationary overriding plate setting¹⁹. An overpressured subslab mantle tends to lift the slab, which may occur in concert with hydrodynamic slab suction above the slab^{19,46,50} and is particularly reported for rollback of the Nazca slab^{22,51} and of the Farallon slab being overridden by the North American plate⁵². The over-pressure of the subslab mantle drives trench-parallel mantle escape during rollback^{43,51} and may even lead to slab stagnation in case of laterally wide slabs for which slowed removal of subslab mantle has become the rollback inhibiting process⁴³.*
”

The discussion of flat slab removal says that tears that ended earlier flat slabs have now disappeared into the deeper upper mantle (Lines 128-130). It would be helpful to have a more detailed discussion about observations that support the proposed model, such as tomographic images, geological features suggesting slab tears (e.g., adakitic magmas; Hu and Liu, 2016). Right now, this aspect of the model comes across as fairly ambiguous and unsupported.

As for the magmatic evidence of slab tears, it is correct that people have linked the arrest of flat slab formation to the formation of adakites, and that also slab tears have been linked to adakites. But the geochemical literature is not conclusive on this link (e.g. Hu and Liu, 2016). Adakites may also form when asthenosphere wells up below hydrated lithospheric mantle when a flat slab rolls back and steepens, not (necessarily) related to slab tears. So based on the geochemistry, no conclusive link can be made between the magmatism and slab tearing and we refrain from putting such links into our paper: we leave this to specialists in magmatic petrology in future work.

We now refer in the manuscript to Hu and Liu (2016) for prediction of slab tears, and to Antonijeev et al. 2015 & 2016 for observational evidence including for mantle flow through the tear. We refer to Scire et al. (2016) for tomographic evidence for a deep upper mantle slab tear (our interpretation) under the Central Atlas as an example of a slab hole that accommodates rollback. We know of no further evidence for slab tears/holes in the Nazca slab.

We added: *“Subsequent subducting of the slab tears can accommodate mantle throughput from the subslab mantle to the topside⁵² until they disappear in the deep mantle. A possible example of a deep Nazca slab conduit, although of uncertain origin, facilitating such mantle flow can be found in recent tomography⁵⁸ of the Nazca slab under the Bolivian Orocline, Central Andes, where below depths of 600 km an eastward pointing cusp in slab geometry develops that seems associated with a hole in the slab. This deep tear in the slab may perhaps have a geometrical (lateral slab bending) origin as it also occurs as such in recent 3-D modelling of the Nazca subduction⁴⁶. An overview of SKS splitting results{Long:2016fy} shows anomalous and puzzling E-W fast splitting patterns in the subslab mantle⁵⁹ that appear to align with the deep slab hole and for which shear in the mantle flow escaping eastward trough*

the hole, and therefore contributing to rollback of the Central Andes slab, could provide a plausible explanation."

Other minor comments:

- in the abstract, the meaning of "horse-shoe slab geometry" (Line 10) is not clear. I did not understand this until after reading the manuscript.

Changed to 'flat slab geometry'.

- Lines 76-80 – there should be a reference to the initial Russo and Silver studies that discuss slab movement and mantle flow (Science 1994; Geology 1996)

Added

- Figure 2 – what are Stage 1 and Stage 2? Seems like these don't correspond with the Stages in the Figure 3 caption.

We explained what these stages mean. They do not refer to the original Fig. 3. and 'stage 1 and 2' are omitted from the caption. We further clarified the caption.

- also on Figure 2 - I don't understand the vertical blue line. The caption says that it is a locally thickened part of the plate. The figure suggests this is a very thin region, but oceanic plateaus are typically hundreds of km wide (Line 99 suggests a width of 1000 km).

This was indeed confusing. Whether or not this lithosphere is thickened or not is irrelevant for the concept that this cartoon aims to display, and has been deleted. It is simply a marker for the lithosphere that was at the trench at the onset of overriding plate advance. Corrected in the caption.

Reviewer #2 (Remarks to the Author):

Review of "South-America plate advance and forced Andean trench retreat as drivers for transient flat subduction episodes"

This manuscript presents a reconstruction of the South American subduction zone for the last ~50 Myr and discusses a conceptual model to provide new insight into the process of flat slab subduction, a process in which a significant portion (several hundred km) of the subducted slab lies (sub-)horizontally below the leading edge of the overriding plate. The reconstruction presented in Fig. 1 and the cartoons presented in Fig. 2 very nicely illustrate and explain (both for South America and in a generic sense) how flat slabs can grow in a kinematic framework, where oceanward retreat of the trench is faster than oceanward migration of the slab hinge at depth (i.e. the bending zone that connects the flat slab segment and the deeper (and steeper) portion of the slab).

The paper is very topical and brings important new insight and kinematic constraints to the debate on the origin of flat slab subduction, which is currently a hot topic in the literature. There are a number of (relatively minor) issues and comments that need to be addressed, most importantly related to the significance of

sub-slab overpressure and the choice of “absolute” reference frame (see below). Once these are addressed, I would recommend that the paper is accepted for publication.

Best wishes,

Wouter Schellart

Main comments:

On L115-118 and L122-127 you argue that a buoyancy-driven change (reduction) in subduction dip angle creates an overpressure in the sub-slab mantle region, which then creates and sustains the flat slab segment. However, one would think that a buoyancy-driven reduction in slab dip angle would cause an underpressure in the sub-slab domain (as the slab is lifted upward, thereby reducing the pressure in the mantle below the slab). As such, the origin of sub-slab overpressure is not entirely clear and requires some more discussion and explanation. (note also that you write both “overpressure” and “over-pressure”).

The reviewer is correct; we have not correctly phrased this. We have largely rewritten this and clearly separate cause and effect and include more information on the role and cause of sub-slab pressure.

This led to: *“We propound that in the case of a laterally long subduction zone with trench retreat forced by the overriding plate (as for the Nazca and Farallon subduction zones), an over-pressured sub-slab mantle immediately responds to any local decrease in subduction angle, initiated by one or more of previously proposed flat slab triggers^{4,11,15,17,19,21,54}, to help create and maintain a flat slab that is instantaneously underlain by the over-pressured sub-slab mantle and that may impede rollback of flat-slab segments, explaining our primary observation. Importantly, in the case of long subduction under a mantle-stationary overriding plate with large lithosphere thickness, 3-D numerical shows that local flat slab creation due to hydrodynamic slab suction alone is highly transient and short-lived (< 5-6 Myr) because of sinking of slab to either side of the flat portion⁴⁸ or isolated by tearing directly around the flat slab⁵⁵. We explain this as resulting from the absence of an over-pressured sub-slab mantle which otherwise could contribute to longer flat slab support.”*

Reconstruction in Fig. 2c: This reconstruction uses the hotspot reference frame of Doubrovine et al. [2012]. Why do you choose this reference frame? You have to provide some justification in the main text (or refer to a justification that you provide in the SI). At present, you provide no justification for this choice, while it is generally known that different choices of reference frame (e.g. Pacific hotspot of Gripp and Gordon [2002], Indo-Atlantic hotspot of Muller et al. [1992] or O’Neill et al. [2005], global hotspot [e.g. Gordon and Jurdy, 1986; Doubrovine et al., 2012], no-

net-rotation reference frames, etc.) can provide substantially different plate velocities and trench velocities [e.g. Funiello et al., 2008; Schellart et al., 2008; Shephard et al., 2012; Williams et al., 2015]. As such, the choice of your reference frame affects the numerical outcomes of your kinematic reconstruction (e.g. trench migration distance, trench velocity, rollback distance, etc.). You illustrate this in Fig. S5, but this is for three rather similar reference frames. For example, what would be the trench migration and trench velocity in the (very different) Pacific hotspot reference frame of Gripp and Gordon [2002]?

I personally favour Indo-Atlantic hotspot reference frames as, for example, there are indications that the Indo-Atlantic hotspot reference frames (e.g. from O'Neill et al. [2005]) work very well as a subduction zone/deep mantle reference frame, because they are most in agreement in scenarios when surface observations and plate tectonic reconstructions are tied with observed deep mantle slab anomalies [e.g. Schellart and Spakman, 2015]. Additionally, they are most in agreement when combining reference-frame dependent subduction kinematics, dynamic subduction modelling and observed slab geometries [e.g. Schellart, 2011], and agree with V-shaped structures of fracture zone segments linking across spreading ridges in the Atlantic Ocean [Muller and Roest, 1992].

In any case, some discussion and justification is required.

We thank the reviewer for this comment. We have added a section addressing this issue. We have chosen the reference frame based on a combination of factors. First, the frame should cover the entire Cenozoic (which excludes Gripp & Gordon), should not be based on fixed hotspots, because this assumption is demonstrably incorrect (excluding frames prior to the 2005 O'Neill frame, which is the first moving hotspot frame), and should include paleolongitudinal control since South America is moving longitudinally (which excludes true polar wander-corrected paleomagnetic frames). That leaves us with the three moving hotspot reference frames and the slab-fitted TPW corrected pmag frame of van der Meer. All these frames perform well against a set of independent criteria for realistic subduction zone behavior formulated by Schellart et al 2008, and tested by Williams et al 2015. The van der Meer frame, however, is subject to much larger uncertainties in the Cenozoic than the moving hotspot reference frames and is designed particularly for Mesozoic time. Therefore, we restrict ourselves to the moving hotspot reference frames, of which we show the only global frame – Doubrovine et al 2012 – and indicate how using Indo-Atlantic frames differ from this (+/- ~100 km since 50 Ma).

We added to the paper: *“This relationship between the amount of subducted lithosphere since the onset of flat slab formation and the width of the flat slab suggests that the flat slabs may not be directly related to a resistance of the Nazca lithosphere against subduction. We therefore evaluate whether an alternative explanation may be found in the relationship between trench motion, slab bend motion, absolute SAM plate motion, and the observed flat slab by placing our reconstruction in a mantle reference frame. This will determine the amount of trench retreat relative to the mantle since 50 Ma. A recent study³³ tested mantle reference frames against a set of independent geodynamic criteria of realistic subduction zone behaviour³⁴ and found that the global moving hotspot reference frame²⁵, two Indo-Atlantic moving hotspot reference*

frames^{35,36}, and a slab-fitted, true polar wander-corrected paleomagnetic reference frame³⁷ perform best. Because the latter frame is subject to the largest, and poorest defined uncertainties and was designed particularly for Mesozoic absolute plate motions, we use the moving hotspot reference frames to place our reconstruction in absolute plate motion context. The global moving hotspot reference frame²⁵, shows $\sim 1400 \pm 180$ km of westward absolute South American plate motion since 50 Ma. The Indo-Atlantic moving hotspot frame predict a motions that are ~ 100 km more³⁵, or less³⁶ since this time (Fig. 6)."

Other comments:

L20 and L22: "towards the overriding plate" This is a bit ambiguous, as the volcanic front and deformation are already located in the overriding plate. Maybe you could say something like "away from the trench"?

Corrected

L74-75: "the observed flat slab only constitutes the last ~ 3 Myr of subduction." I would replace this with something like "the observed flat slab segment only constitutes the last ~ 3 Myr of subducted oceanic lithosphere." to be more precise.

Changed to 'only constitutes lithosphere that subducted since 3 Ma'.

L82: "... friction would have been highest." This has not much to do with friction coefficients. It would be more accurate to say "... normal and shear stresses at the subduction zone interface would have been highest".

Corrected

L122-127 and elsewhere. One of the general conclusions, that flat slab subduction is caused by both forced trenchward overriding plate motion, which forces trench retreat, and subduction of buoyant features, is essentially the same as that of van Hunen et al. [2002], who base their conclusion on numerical subduction models of progressive subduction in 2D space. I think this work should be cited in the text.

Citation added

L164: "upper mantle" Be more precise: this is the sub-lithospheric upper mantle.

Corrected

L165: "cross section trench-perpendicular" Replace with "trench-perpendicular cross section"

Corrected

L166-167: "the slab bend as the hinge between the (sub-)horizontal and dipping portions of the subducting plate"

This definition would actually indicate the bend at the trench, so you need to be more precise, e.g. "the hinge between the (sub-)horizontal slab segment underlying the overriding plate and the dipping slab segment further down the subduction zone"

Thanks. Corrected.

L173 and L183: Replace "contraction" with "shortening"

Corrected

L199: "Slab healing" is not the right choice of words here, as it would imply that at the place of the slab tear, this tear "heals" and so disappears, making the slab continuous again. This is not what you mean here. You mean that the slab segment with the slab tear progressively subducts to deeper levels and is replaced at shallower levels by slab material that is continuous.

Anyway, I suggest to rephrase to be more precise and avoid possible confusion.

Rephrased according to this suggestion.

In Fig. 3, for stages $t=1$, $t=2$ and $t=3$, the curvature of the slab edges (convex towards the mantle wedge side) is the wrong way. Toroidal mantle return flow at the lateral edges of a wide slab (and narrower slabs for that matter) forces these lateral slab edges to attain a curvature that is concave towards the mantle wedge side. See for example Fig. 4 in Schellart et al. [2007] and Fig. 4 in Morra et al. [2006].

Corrected, thanks for spotting.

Reviewer 3

South-America plate advance and forced Andean trench retreat as drivers for transient flat subduction episodes

General Review Questions:

1. *Who will be interested in reading the paper, and why?*

This is an exciting paper that will be of great interest to the broader geoscience community, including the subdisciplines of geology, structural geology, tectonics, sedimentology and stratigraphy, geophysics (paleomagnetism and seismology), and geodynamics. The paper would reach a broad audience because of the use of a range of data types and the application of these data to addressing the general and not fully understood tectonic process of flat slab subduction.

We thank the reviewer for these positive comments.

2. *What are the main claims of the paper and how significant are they?*

The paper claims that it can connect the width of the portion of the subducting plate that is undergoing flat slab subduction to the relative motion between the trench and the rollback of the distal bend in the flat slab. The paper suggests the mechanism to maintain the flat slab is the overpressured sub-slab mantle, that has been trapped due to a particular 3D slab configuration that prevents the subslab mantle from escaping. Eventually, the slab tears and the overpressured mantle can escape, reducing subslab support on the slab and allowing the slab to steepen. Thus, this is a time-dependent phenomenon that can explain the changes in flat slab geometry through time.

In addition, the paper presents a synthesis of geologic data used to construct estimates of tectonic shortening along the South American subduction margin. The shortening estimates are compared to that by previous studies and show an along strike variation in shortening (see supplementary figure). The shortening data are used in the tectonic reconstruction of the upper plate. In addition, the data may be used to constrain where the distal bend of the slab was located in the subsurface, which is in turn used to constrain the width of the flat slab segment through time. However, this is not clear. Uplift constraints and volcanism are also used to constrain the paleo-geometry and timing of changes in slab morphology.

The reviewer here misunderstood part of our analysis. The reconstruction we made cannot constrain the width of flat slabs in the geological past. For that, we rely on conclusions drawn in earlier publications, e.g. by Ramos and co-workers, who infer locations and dimensions of flat slabs from data on volcanism and uplift. What we show for the example of the present-day flat slabs is that the current width of the flat slabs coincides with the amount of westward absolute trench motion of the Andean trench since the 12 Ma onset of flat slab formation. From this, we draw the conclusion that the slab bend has been essentially stationary in the mantle, and roll-back was apparently impeded. We infer that previous stages of flat slab subduction, inferred from magmatic and structural geological records, can be explained in a similar way.

The geodynamic implications are very significant, and put forward a newly formed perspective on flat slab subduction processes very closely tied to observational data. The data compilation presented in the supplemental information is quite extensive. It by itself will likely be used in future studies.

We thank the reviewer for this positive comment.

3. Is the paper likely to be one of the five most significant papers published in the discipline this year?

It is not clear if it will be in the top 5, but it certainly will raise quite an interest and probably direct multiple new lines of inquiry in many subdisciplines in geoscience. In other words, it has the potential to be a very significant paper and have a large impact on the field.

We thank the reviewer for this positive comment.

4. How does the paper stand out from others in its field?

The paper stands out in terms of the synthesis of data and the integration of geologic observational data into a tectonic reconstruction as well as to a geodynamic framework that is closely tied to recent results from studies in computational geodynamics.

We thank the reviewer for this positive comment.

5. Are the claims novel? If not, which published papers compromise novelty?

The geodynamic mechanism has been recently suggested in several papers cited in the paper. However, the other papers do not make the connections to the geologic

data and the tectonic reconstructions in the way and the extent that is presented in this paper.

We thank the reviewer for this positive comment. As noted by the first reviewer, we aim to bring the problem back to its essence, and we feel that in that lies the clarity and novelty of our paper.

6. Are the claims convincing? If not, what further evidence is needed?

There is likely sufficient evidence to support the claims. However, in places the paper could benefit from clarification of what the specific evidence are and how they are used in the paper.

For example, are the thin-skinned deformation constraints being applied to constrain wholesale lithospheric shortening and previous plate length? If so, the claims could be strengthened by clarification of how thin-skinned crustal deformation is used to constrain deformation of the entire plate (mantle lithosphere as well as crust).

Because the Andes form by shortening of the overriding plate, upper crustal shortening must be equal to lower crustal and lithospheric mantle shortening. Upper crustal shortening is rarely (if ever) distributed in the same way as upper mantle shortening. Upper mantle shortening is commonly facilitated by either abrupt (delamination) or systematic lithosphere removal (or 'ablative subduction' as it was called by Pope and Willett 1998). We have cited Pope and Willett 1998 to support our assumption that the upper crustal shortening is representative for entire plate shortening.

In addition, what exactly is used to identify where the slab was flat and to identify the distance from the trench at which the bend in the flat slab occurred in the past? For example, recent geodynamic models show the distal bend in a flat slab often correlates with a basin in the upper plate, e.g., Jadamec et al. (EPSL 2013) and Eakin et al. (EPSL 2014).

For our analysis, we have used the widely used estimates of the onset of flat slab subduction, which are based on geological observations including volcanological, uplift, and shortening evolution of the orogen (Ramos and Folguera 2009; Jordan et al, 1983). These have estimated a 12-11 Ma onset of flat slab subduction. We further use the present-day width of the flat slabs of ~250 km imaged seismologically, and used the plate reconstruction in an absolute plate motion frame to estimate the net amount of trench advance driven by westward motion of South America, corrected for Andean shortening. It turns out that that the amount of westward trench advance since the 12 Ma onset of flat slab subduction is equal to the modern width of the flat slabs, from which we conclude that the bend bounding the flat slabs in the east not rolled back in the last 12 Ma. In other words, our conclusions are independent from circumstantial observations such as the ones suggested here by the reviewer. Our reconstruction can hence be used to test hypotheses such as those put forward by Jadamec et al 2013, rather than use those to constrain our reconstruction.

The relative motions and importance of reference frames are emphasized, but they could be better logically connected within the text. Figures 1 and 2 are the opportunity to do this, but could be better presented. Clarify why the different reference frames are used for Figure 1b,c.

We have done this already in response to the comments of Reviewer 2.

Figure 1 in general is hard to follow. In the reconstructions, is the trench geometry fixed?

This comment refers to the new Figure 5. Both figures clearly indicate what is fixed: in Figure 5A we have fixed the Nazca plate, because we want to show how much of the Nazca plate must have been lost to subduction since 12 Ma. In Figure 5B, we place the reconstruction in the Doubrovine et al mantle reference frame to show absolute trench migration.

Does it curve through time as in Schellart et al. (Nature, 2007), and what are the implications of a time evolving plate boundary shape, e.g. Moresi et al. (Nature, 2015), on the reconstruction length estimates?

Our reconstruction follows the kinematic constraints obtained from the field, and is thus not necessarily the same as those in numerical models such as those in Schellart et al 2007. We use our reconstruction as boundary condition for dynamic analysis, so it is the reconstruction length estimates that constrain the time evolving plate boundary shape, rather than the other way around. Our reconstruction shows how the plate boundary shape evolved over time, which is generally consistent with those in Schellart and Moresi.

Clarify the velocity diagrams in Figure 2, as well as the subduction modes and their labeling.

We have done this already in response to the comments of Reviewer 2.

What is Figure 3 based on? Is this from a geodynamic model result from the authors or elsewhere in the literature? Is it from the flat slab width determined from geologic estimates through time?

As we quite clearly explain in the text, this cartoon is a conceptual evolution of flat slabs. The flat slab width here is schematic, and irrelevant for the concept. This is also clearly indicated in the caption of Figure 3 (now Figure 7)

Figure 3 is a good attempt to use simplicity to convey the concepts, but the parts are not clearly labeled. For example, what is the gray swiggly band in the left diagram? Is that the more buoyant body?

That may indeed be a more buoyant body, as we now indicated in the caption, but we leave the option open that there are other triggering mechanisms. The main point of our paper is that the trigger only needs to develop an incipient flat slab: the development of the tunnel and overpressure in the trapped mantle will then lead to inhibited roll-back, leading to flat slabs.

Also, according to Gutscher et al. (Tectonics, 2000), for example, the length of the modern flat slab segments beneath South America can be quite variable in some regions are greater than 400 km. This longer length is not really addressed. Can the authors comment?

We have based our flat slab length estimates on the latest state-of-the-art of seismological observations, e.g. Hayes et al., 2012, which indicates a 200-300 km wide length. This appears to be the current consensus. With our Gplates reconstruction that we have added as supplementary files, the reader can play with alternative flat slab length scenarios.

7. Are there other experiments or work that would strengthen the paper further?

There is a large data compilation. How is it used could be better clarified. Including a geodynamic model result in the supplemental figure would be useful to demonstrate the subslab pressure, and the time evolution of the subslab pressure changes and slab dip changes, if the authors have a figure from previous work.

Unfortunately this does not exist in the literature. We have now included more reference to modeling papers mentioning effects of sub-slab pressure (earlier new paragraphs). Basically, we articulate a new type of forcing assisting in flat slab creation and particularly for maintaining flat slabs over extended periods.

8. How much would further work improve it, and how difficult would this be? Would it take a long time?

I think it would be in the scope of a minor to major revision. Not every reader may agree with the conclusions, but if the assumptions are more clearly laid out, then the reader could better assess the results.

We have clarified the text where requested

9. Are the claims appropriately discussed in the context of previous literature?

There is actually a very good overview of the relevant literature. The authors have done a nice job on this, particularly so, with the breadth of the paper.

In addition to the papers above, it would be useful to include (a) Schellart (JGR, 2004) in the discussion of toroidal and poloidal flow (lines 77-78)

Added

and (b) Jadamec and Billen (Nature, 2010) and Stadler et al. (Science, 2010) in the discussion in lines 78-79, as the upper mantle viscosity plays a major role in determining the mantle flow speed.

We found the Jadamec and Billen paper most appropriate for illustration of toroidal flow.

In addition, in linking the mantle flow to upper plate shortening and deformation, these two papers are relevant (Capitanio et al. (Nature, 2011) and Schellart and Moresi (JGR 2013)).

We evaluate that the Capitanio et al. 2011 reference is most appropriate.

This paragraph in general, lines 76-84, is unclear.

We revised this as follows "We now explore a dynamic explanation for the stagnation of the slab bend relative to the mantle during flat slab formation. A thousand kilometres of trench retreat and without forming flat slabs of this length requires slab rollback. When viewed in a mantle reference frame, slab rollback requires that mantle material is removed from below the slab. This occurs through toroidal flow around the edges of, or through tears or holes in the slab, and poloidal flow around its base if possible³⁸⁻⁴². The mantle flow speed and lateral length of the flow path in the sub-slab mantle limits the rollback speed and facilitates slab stagnation and potential flat slabs for laterally-long subduction zones⁴³. Slab stagnation may explain why most shortening, averaged over the last 50 Ma, occurred in the central segment of the Andes²⁹, a proposed potential rollback "stagnation point"⁴⁴ based on seismological inferences of trench-parallel mantle flow in the sub-slab mantle. Assuming a contiguous 7000 km slab below the Andes, rollback would have been slowest in this central segment, and normal and shear stresses at the subduction interface between the advancing SAM and the downgoing Nazca plate would have been highest^{43,45}. Thus flat slab formation is expected in the 50 Ma subduction history of the central Andes. However, the current slab geometry, with two flat slabs segments separated by a dipping slab that has been able to roll back rather than one flat segment in the central part of the Andean subduction zone suggests that the triggers and maintenance of flat slabs require additional mechanisms"

Mason et al. (Tectonophysics, 2010) is relevant for lines 119-120.

Reference included (twice)

For line 119, is the tearing of the edge of the slab a lateral edge as in Jadamec and Billen (Nature, 2010) or at the base of the slab (Arrial and Billen)?

Neither. It is as in Mason et al. 2010, which we included as a reference.

In lines 33-36, it would be useful to include other models that show the effect of the upper plate on flat slab subduction, that suggest additional mechanisms, e.g., Sharples et al. (JGR, 2014) and Rodríguez-González et al. (EPSL, 2014).

The flat slabs in Sharples et al. 2014 (their fig. 6) are not really flat slabs. We included the second reference.

10. *If the manuscript is unacceptable, is the study sufficiently promising to encourage the authors to resubmit?*

This is a very interesting paper, and I recommend major revision.

11. *If the manuscript is unacceptable but promising, what specific work is needed to make it acceptable?*

Additional Questions to Consider:

1. *Is the manuscript clearly written? If not, how could it be made more clear or accessible to nonspecialists?*

The paper is clearly written. There could be some tightening in terms of the logic in linking how the observations, geodynamic modeling results, and choice of reference frames all lead to the main result.

We have clarified the text where requested

2. Would readers outside the discipline benefit from a schematic of the main result to accompany publication?

Yes, but I think this is what Figure 3 is aiming to do. Revision of Figure 3 could achieve this.

We have clarified Figure 3 where requested above

3. Could the manuscript be shortened? (Because of pressure on space in our printed pages we aim to publish manuscripts as short as is consistent with a persuasive message.)

Yes, the paper could be tightened a little in terms of streamlining the logic. This would likely not save more than 1 paragraph though.

Following the request of the editor to include the main findings listed in the original Supplementary Information, and in response to requests by the reviewers, the text has been increased in length, but well within the limits set by the journal format.

4. Should the authors be asked to provide supplementary methods or data to accompany the paper online? (Such data might include source code for modelling studies, detailed experimental protocols or mathematical derivations.)

More information on the tectonic reconstructions in terms of the upper plate length and subducting plate paleo-geometry would strengthen the paper.

We have addressed these comments above

5. Have the authors done themselves justice without overselling their claims?

I think so. Addressing the comments elsewhere in the review and including the broader scope of geodynamic modeling studies would do the paper more justice, but expanding the perspective.

See rebuttal to earlier comments

6. Have they been fair in their treatment of previous literature?

Yes, the authors have done a good job. Including several additional papers would give a more balanced discussion.

We have included the suggested literature above

7. Have they provided sufficient methodological detail that the experiments could be reproduced?

Yes, but again, more detail on the reconstructions and what exactly was used, and what points were measured to obtain slab length, would be strengthen the paper.

See responses to comments above

8. Is the statistical analysis of the data sound, and does it conform to the journal's guidelines?

There are no error bars on the individual shortening estimates, but this may be hard to constrain. In comparing the different data sets, it is not clear how to

systematically weigh the studies if there were to be significant error ranges between studies.

We have added a sentence in the text where referred to the new Figure 4 (which came from the Supplementary Information), in which we provide an uncertainty estimate by comparing our reconstruction to previous estimates. Our estimates are a maximum value, and provide the minimum rate of trench retreat.

It would be useful to show a plot of the slab length through time and what data were used to constrain that length, and for what parts of the subduction zone.

Figure 1 may be trying to do this, but what geologic data from the surface determines the location of the bend in the slab at depth?

We have explained our procedure earlier in this rebuttal: we did not constrain the growth rate of the flat slabs in any further detail than comparing today's length observed seismologically with the onset of growth estimated previously from geological records.

9. Are the reagents generally available?

I don't the reagents question applies to this paper. Chemical experiments were not being conducted.

10. Are there any special ethical concerns arising from the use of human or other animal subjects?

No, there are no special ethical concerns regarding human or other animal subjects.

Reviewers' Comments:

Reviewer #1 (Remarks to the Author)

I have read the revised version of this manuscript and I find that the authors have made numerous changes to address comments raised in my review and the other two reviews. I believe that these changes have clarified their arguments and have made the manuscript stronger. The readability of the paper has been improved by moving figures/material from the Supplementary Information to the main text.

I am happy to recommend publication of this manuscript. As mentioned in my earlier review, this is a nice study that presents a clear hypothesis for the evolution of flat slabs. While, it is based on mechanisms that have been explored in earlier work, and I agree with the authors' rebuttal that this work is novel in that it emphasizes the role of high sub-slab pressures associated with plate motions.

I appreciate that more details have been added about the observational evidence for removal of flat slab segments (Lines 211-222). I do feel that this part of the story is more speculative than the proposed model for development of flat slabs. However, I do not think that any further revisions are needed. To date there have been very few geodynamic modeling studies of slab re-steepening. This manuscript puts forward a conceptual idea that appears to be supported by observations, and this will undoubtedly spur modeling studies to test it. Therefore, this should be considered a strength of the manuscript.

I do not have any significant comments for further revisions, but do have some very minor suggestions of typos/grammar:

Line 51 - "lithosphere INCREASE the propensity" (no s)

Line 59 - "50 Ma" rather than "50 Myr"

Line 95 - no capital on "south"

Line 97 - "the largest NUMBER of balanced"

Line 113 - "and the width" - should this be "length"? (confusing whether it refers to margin-parallel or margin-normal dimensions)

Line 141 - "the edges of" - of what? is there a word missing?

Line 160 - "combined with and a" - remove word "and"?

double commas on Lines 57, 75

Figure 4 - on top axis, indicate it is degrees south?

Reviewer #2 (Remarks to the Author)

Review of revised manuscript "South-America plate advance and forced Andean trench retreat as drivers for transient flat subduction episodes"

By Schepers et al.

The authors have made a considerable effort to improve their manuscript and to accommodate the comments from the reviewers. The reasoning, argumentation, rationale, clarity and illustrations of the manuscript have all improved considerable, and so the manuscript reads very well and is easy to follow. I am satisfied as to how my own comments have been incorporated. I only have two very minor comments left (see below), and so would recommend that the paper is accepted for publication.

Best wishes,

Wouter Schellart

On line 150, I don't think reference 45 is appropriate here, as these authors ascribe the slower rollback and the higher stresses in the centre entirely to the thickness of the overriding plate (which they incorporated a priori in their numerical model and is not an actual outcome of their model). These authors do not ascribe it to the large width (trench parallel extent) of the subduction zone, which is what you ascribe it to and what reference 43 proposed.

Check for British versus American spelling, e.g. L208 "modeled".

)

Reviewer #3

General Comments:

I have read through the revised manuscript and the response to reviewers. I am satisfied with the revisions to the manuscript and recommend it for publication. Below are very minor comments, mostly to aid in clarity in the revision.

Lines 29-30: A suggestion here is to replace “explaining the transient character of flat slabs” with something like “providing a mechanism for the transient character of flat slabs”. This is because geodynamic models have shown that flat slabs can be transient without tearing. For example, depending on the density ratio and lateral extent of a buoyant feature in the subducting plate, a slab can become flat and then have the buoyant feature subduct, without tearing occurring (e.g., Arrial and Billen, 2013). As written, it implies the paper explains the transient nature of all flat slabs.

Lines 37-38: There is still a mantle wedge. It is just located more distal to the trench and at a deeper depth.

Line 65: “rollback of the slab bend” is that what the authors mean? It would be useful to explicitly specify “rollback of the slab bend” or indicate very clearly that that is what is meant when rollback is referred to in the paper, to avoid confusion with previous use of the term in the literature.

Line 75: extra comma

Lines 84-85: Clarify the assumption.

Lines 105-108: Clarify. Is it that 1000 km has been subducted and/or that there was a flat slab length of 1000 km. It is unclear as written.

Lines 112-113: Clarify assumptions in argument.

Lines 163-164: Has a 1000 km long aseismic ridge been ruled out by independent observations? Add citation for why this is unlikely.

Line 178: Why is “in and advancing” in green font?

Lines 218-222: Align in what way? Be more specific. Also, at what depth? How far below 600 km? Is this below the transition zone? If diffusion creep and not dislocation creep dominates in the lower mantle, then would the shearing produce LPO in the upper most lower mantle?

Figure 1: Is the orange-red color SA? If so, move SA to the left. For the long green-brown domain, is that both PC and PAC?

Figure 2: This is better, but still unclear. Both Modes 2 and Mode 3 form a flat slab. Mode 2 is not just a “partially flat slab”. It is a flat slab, just a short one. Clarify the labeling and conceptual description. For Mode 3 in the diagram and caption – is “flat slab formation rate” formally defined in the manuscript?

Figure 3: Add the letters to the geomorphic zones on Figure 3. One simply cannot refer to the caption for Figure 1 to find the provinces in Figure 3, because the caption to Figure 1 defines regions by acronyms not colors. It is a minor point, but will make the figure more useful. Also, The caption has a typo. Refer to Figure 1 (not Figure 3).

In the uploaded document, Figures 5 and 6 are switched in order, so are out of sync with the respective captions.

Figure 5: Clarify the significance of “Dipping slab: ~ 1000 km roll-back” and “Flat slab: ~750 km roll-back” in the figure inset.

Figure 7: Why is there trench-parallel sub-slab mantle flow at time 0? Is this related to the toroidal flow? Otherwise, what would causing the pressure gradient to drive the sub-slab mantle flow, for a subduction zone with a straight trench with a constant slab dip? Also, specify “sub-slab mantle flow”.

Additional: This explanation in the rebuttal “What we show for the example of the present-day flat slabs is that the current width of the flat slabs coincides with the amount of westward absolute trench motion of the Andean trench since the 12 Ma onset of flat slab formation. From this, we draw the conclusion that the slab bend has been essentially stationary in the mantle, and roll-back was apparently impeded. We infer that previous stages of flat slab subduction, inferred from magmatic and structural geological records, can be explained in a similar way.” summarized the points in a very straightforward way. It would be useful to actually add this text in the manuscript.

REVIEWERS' COMMENTS:

Reviewer #1 (Remarks to the Author):

I have read the revised version of this manuscript and I find that the authors have made numerous changes to address comments raised in my review and the other two reviews. I believe that these changes have clarified their arguments and have made the manuscript stronger. The readability of the paper has been improved by moving figures/material from the Supplementary Information to the main text.

I am happy to recommend publication of this manuscript. As mentioned in my earlier review, this is a nice study that presents a clear hypothesis for the evolution of flat slabs. While, it is based on mechanisms that have been explored in earlier work, and I agree with the authors' rebuttal that this work is novel in that it emphasizes the role of high sub-slab pressures associated with plate motions.

I appreciate that more details have been added about the observational evidence for removal of flat slab segments (Lines 211-222). I do feel that this part of the story is more speculative than the proposed model for development of flat slabs. However, I do not think that any further revisions are needed. To date there have been very few geodynamic modeling studies of slab re-steepening. This manuscript puts forward a conceptual idea that appears to be supported by observations, and this will undoubtedly spur modeling studies to test it. Therefore, this should be considered a strength of the manuscript.

I do not have any significant comments for further revisions, but do have some very minor suggestions of typos/grammar:

Line 51 - "lithosphere INCREASE the propensity" (no s)

Corrected

Line 59 - "50 Ma" rather than "50 Myr"

Corrected, throughout text

Line 95 - no capital on "south"

Corrected

Line 97 - "the largest NUMBER of balanced"

Corrected

Line 113 - "and the width" - should this be "length"? (confusing whether it refers to margin-parallel or margin-normal dimensions)

Corrected, also in previous sentence. Sentence now reads: *This relationship between the **length** of subducted lithosphere since the onset of flat slab formation and the **length** of the flat slab suggests that the flat slabs may not be directly related to a resistance of the Nazca lithosphere against subduction*

Line 141 - "the edges of" - of what? is there a word missing?
Slab edges. Corrected.

Line 160 - "combined with and a" - remove word "and"?
Corrected

double commas on Lines 57, 75
Corrected

Figure 4 - on top axis, indicate it is degrees south?
Done

Reviewer #2 (Remarks to the Author):

Review of revised manuscript "South-America plate advance and forced Andean trench retreat as drivers for transient flat subduction episodes"

By Schepers et al.

The authors have made a considerable effort to improve their manuscript and to accommodate the comments from the reviewers. The reasoning, argumentation, rationale, clarity and illustrations of the manuscript have all improved considerably, and so the manuscript reads very well and is easy to follow. I am satisfied as to how my own comments have been incorporated. I only have two very minor comments left (see below), and so would recommend that the paper is accepted for publication.

Best wishes,

Wouter Schellart

On line 150, I don't think reference 45 is appropriate here, as these authors ascribe the slower rollback and the higher stresses in the centre entirely to the thickness of the overriding plate (which they incorporated a priori in their numerical model and is not an actual outcome of their model). These authors do not ascribe it to the large width (trench parallel extent) of the subduction zone, which is what you ascribe it to and what reference 43 proposed.

Reference 45 has been removed.

Check for British versus American spelling, e.g. L208 "modeled".
Corrected

Reviewer 3:

General Comments:

I have read through the revised manuscript and the response to reviewers. I am satisfied with the revisions to the manuscript and recommend it for publication. Below are very minor comments, mostly to aid in clarity in the revision.

Lines 29-30: A suggestion here is to replace “explaining the transient character of flat slabs” with something like “providing a mechanism for the transient character of flat slabs”. This is because geodynamic models have shown that flat slabs can be transient without tearing. For example, depending on the density ratio and lateral extent of a buoyant feature in the subducting plate, a slab can become flat and then have the buoyant feature subduct, without tearing occurring (e.g, Arrial and Billen, 2013). As written, it implies the paper explains the transient nature of all flat slabs.

Corrected as suggested

Lines 37-38: There is still a mantle wedge. It is just located more distal to the trench and at a deeper depth.

Remark ‘due to elimination of the mantle wedge’ has been deleted, sentence now reads ‘extinction of the magmatic arc’, without specification about dynamic causes.

Line 65: “rollback of the slab bend” is that what the authors mean? It would be useful to explicitly specify “rollback of the slab bend” or indicate very clearly that that is what is meant when rollback is referred to in the paper, to avoid confusion with previous use of the term in the literature.

We now defined roll-back: *‘rollback, which we define as the retreat of the slab bend relative to the mantle’*

Line 75: extra comma

Corrected

Lines 84-85: Clarify the assumption.

We added: *Our restoration uses published maximum estimates of Andean shortening, which leads to a minimum estimate of trench retreat **and hence reconstructs the minimum amount of roll-back required to avoid the formation of flat slabs.***

Lines 105-108: Clarify. Is it that 1000 km has been subducted and/or that there was a flat slab length of 1000 km. It is unclear as written.

Rephrased to clarify: *Combining the shortening reconstruction of the Andes with the South America-Antarctica-Pacific-Nazca plate circuit²⁶ we arrive at the important result that ~1000 km of Nazca plate has passed the trench and subducted since the 12 Ma onset of flat slab subduction (Fig. 5), **whereas the modern the flat slabs are only ~200-300 km long.***

Lines 112-113: Clarify assumptions in argument.

We don't see where we make assumptions here, we just explain a straightforward kinematic principle.

Lines 163-164: Has a 1000 km long aseismic ridge been ruled out by independent observations? Add citation for why this is unlikely.

Rephrased, since this is not necessarily unlikely. It is just a requirement. *A transition delay of ~3 Myr would explain the ~250 km long Pampean flat slab, provided that the subducted aseismic ridge was at least ~1000 km long*

Line 178: Why is "in and advancing" in green font?

No reason, corrected.

Lines 218-222: Align in what way? Be more specific. Also, at what depth? How far below 600 km? Is this below the transition zone? If diffusion creep and not dislocation creep dominates in the lower mantle, then would the shearing produce LPO in the upper most lower mantle?

We changed the sentence: *"An overview of SKS splitting results{Long:2016fy} shows anomalous and puzzling E-W fast splitting patterns in the subslab mantle⁵⁹ that appear to align with the deep slab hole and for which shear in the mantle flow escaping eastward through the hole, and therefore contributing to rollback of the Central Andes slab, could provide a plausible explanation."*
into

"An overview of SKS splitting results{Long:2016fy} shows anomalous and puzzling E-W fast splitting patterns in the subslab mantle⁵⁹ in a narrow zone (~150 km in N-S extent) that geographically aligns in E-W direction with the deep slab hole. Shear deformation associated with upper mantle flow toward the hole and escaping eastward through the hole could provide a plausible explanation for the observed splitting observations while the flow itself would release subslab pressure and contribute to local rollback of the Central Andes slab"

Figure 1: Is the orange-red color SA? If so, move SA to the left. For the long green-brown domain, is that both PC and PAC?

Yes and yes. Corrected and clarified in figure.

Figure 2: This is better, but still unclear. Both Modes 2 and Mode 3 form a flat slab. Mode 2 is not just a "partially flat slab". It is a flat slab, just a short one. Clarify the labeling and conceptual description. For Mode 3 in the diagram and caption – is "flat slab formation rate" formally defined in the manuscript?

Corrected the figure. Mode 2 and 3 were essentially the same, with Mode 3 having a mantle-stationary slab as in the Andes according to our reconstruction. But to explain the kinematic principle, Mode 3 is unnecessary and has been deleted from the Figure. 'Mode 1 and Mode 2' have also been deleted, and are replaced by simply A and B. Caption updated.

Figure 3: Add the letters to the geomorphic zones on Figure 3. One simply cannot refer to the caption for Figure 1 to find the provinces in Figure 3, because the caption to Figure 1 defines regions by acronyms not colors. It is a minor point, but will make the figure more useful. Also, The caption has a typo. Refer to Figure 1 (not Figure 3).

Corrected and added

In the uploaded document, Figures 5 and 6 are switched in order, so are out of sync with the respective captions.

Corrected, thanks for spotting

Figure 5: Clarify the significance of “Dipping slab: ~ 1000 km roll-back” and “Flat slab: ~750 km roll-back” in the figure inset.

Rephrased: *Dipping slab segments underwent 1000 km roll-back since 50 Ma; flat slabs segments underwent 750 km roll-back since 50 Ma and no roll-back since 12 Ma*

Figure 7: Why is there trench-parallel sub-slab mantle flow at time 0? Is this related to the toroidal flow? Otherwise, what would causing the pressure gradient to drive the sub-slab mantle flow, for a subduction zone with a straight trench with a constant slab dip?

Yes, it is related to the toroidal flow. The time 0 panel is intended as an initial schematic stage in which there is rollback (indicated), hence trench-parallel sub-slab mantle escape flow (indicated) and associated toroidal escape flow around slab edges at time 0 (indicated).

Also, specify “subslab mantle flow”.

Done

Additional: This explanation in the rebuttal “What we show for the example of the present-day flat slabs is that the current width of the flat slabs coincides with the amount of westward absolute trench motion of the Andean trench since the 12 Ma onset of flat slab formation. From this, we draw the conclusion that the slab bend has been essentially stationary in the mantle, and roll-back was apparently impeded. We infer that previous stages of flat slab subduction, inferred from magmatic and structural geological records, can be explained in a similar way.” summarized the points in a very straightforward way. It would be useful to actually add this text in the manuscript.

Thanks for the suggestion. We added to the last part of the paper a modified version of this explanation: *Our kinematic analysis provides first-order constraints on the long-debated formation of the Andean subduction zone flat slab segments: we show for the example of the present-day flat slabs that their length coincides with the amount of westward absolute Andean trench motion since the 12 Ma onset of flat slab formation. From this, we draw the conclusion that the slab bend has been essentially stationary in the mantle, and rollback was apparently impeded. We suggest that previous stages of*

flat slab subduction below the Andes, inferred from magmatic and structural records⁹, can be explained in a similar way.